# Deep Learning with Kernels through RKHM and the Perron–Frobenius Operator

**Yuka Hashimoto**
NTT Network Service Systems Laboratories /
RIKEN AIP,
Tokyo, Japan
yuka.hashimoto@ntt.com

**Masahiro Ikeda**
RIKEN AIP / Keio University,
Tokyo, Japan
masahiro.ikeda@riken.jp

**Hachem Kadri**
Aix-Marseille University, CNRS, LIS,
Marseille, France
hachem.kadri@lis-lab.fr

## Abstract

Reproducing kernel Hilbert $C^*$-module (RKHM) is a generalization of reproducing kernel Hilbert space (RKHS) by means of $C^*$-algebra, and the Perron–Frobenius operator is a linear operator related to the composition of functions. Combining these two concepts, we present deep RKHM, a deep learning framework for kernel methods. We derive a new Rademacher generalization bound in this setting and provide a theoretical interpretation of benign overfitting by means of Perron–Frobenius operators. By virtue of $C^*$-algebra, the dependency of the bound on output dimension is milder than existing bounds. We show that $C^*$-algebra is a suitable tool for deep learning with kernels, enabling us to take advantage of the product structure of operators and to provide a clear connection with convolutional neural networks. Our theoretical analysis provides a new lens through which one can design and analyze deep kernel methods.

## 1 Introduction

Kernel methods and deep neural networks are two major topics in machine learning. Originally, they had been investigated independently. However, their interactions have been researched recently. One important perspective is deep kernel learning [1, 2, 3]. In this framework, we construct a function with the composition of functions in RKHSs, which is learned by given training data. Representer theorems were shown for deep kernel methods, which guarantee the representation of solutions of minimization problems only with given training data [4, 5]. We can combine the flexibility of deep neural networks with the representation power and solid theoretical understanding of kernel methods. Other important perspectives are neural tangent kernel [6, 7] and convolutional kernel [8], which enable us to understand neural networks using the theory of kernel methods. In addition, Bietti et al. [9] proposed a regularization of deep neural network through a kernel perspective.

The generalization properties of kernel methods and deep neural networks have been investigated. One typical technique for bounding generalization errors is to use the Rademacher complexity [10, 11]. For kernel methods, generalization bounds based on the Rademacher complexity can be derived by the reproducing property. Bounds for deep kernel methods and vector-valued RKHSs (vvRKHSs) were also derived [5, 12, 13]. Table 1 shows existing Rademacher generalization bounds for kernel methods. Generalization bounds for deep neural networks have also been actively studied [14, 15, 16, 17, 18, 19]. Recently, analyzing the generalization property using the concept of benign overfitting

| Reproducing space | Output dimension | Shallow | Deep |
|---|---|---|---|
| RKHS | 1 | $O(\sqrt{1/n})$ [10] | $O(A^L\sqrt{1/n})$ |
| vvRKHS | $d$ | $O(\sqrt{d/n})$ [12, 13] | $O(A^L\sqrt{d/n})$ [5] |
| RKHM (existing) | $d$ | $O(\sqrt{d/n})$ [27] | – |
| RKHM (ours) | $d$ | $O(d^{1/4}/\sqrt{n})$ | $O(B^L d^{1/4}\sqrt{n})$ |

Table 1: Existing generalization bounds for kernel methods based on the Rademacher complexity and our bound ($n$: sample size, $A$: Lipschitz constant regarding the positive definite kernel, $B$: The norm of the Perron–Frobenius operator)

has emerged [20, 21]. Unlike the classical interpretation of overfitting, which is called catastrophic overfitting, it explains the phenomenon that the model fits both training and test data. For kernel regression, it has been shown that the type of overfitting can be described by an integral operator associated with the kernel [22].

In this paper, we propose deep RKHM to make the deep kernel methods more powerful. RKHM is a generalization of RKHS by means of $C^*$-algebra [23, 24, 25], where $C^*$-algebra is a generalization of the space of complex numbers, regarded as space of operators. We focus on the $C^*$-algebra of matrices in this paper. Applications of RKHMs to kernel methods have been investigated recently [26, 27]. We generalize the concept of deep kernel learning to RKHM, which constructs a map from an operator to an operator as the composition of functions in RKHMs. The product structure of operators induces interactions among elements of matrices, which enables us to capture relations between data components. Then, we derive a generalization bound of the proposed deep RKHM. By virtue of $C^*$-algebras, we can use the operator norm, which alleviates the dependency of the generalization bound on the output dimension.

We also use Perron–Frobenius operators, which are linear operators describing the composition of functions and have been applied to analyzing dynamical systems [28, 29, 30, 31], to derive the bound. The compositions in the deep RKHM are effectively analyzed by the Perron–Frobenius operators.

$C^*$-algebra and Perron–Frobenius operator are powerful tools that provide connections of the proposed deep RKHM with existing studies. Since the norm of the Perron–Frobenius operator is described by the Gram matrix associated with the kernel, our bound shows a connection of the deep RKHM with benign overfitting. In addition, the product structure of a $C^*$-algebra enables us to provide a connection between the deep RKHMs and convolutional neural networks (CNNs).

Our main contributions are as follows.
- We propose deep RKHM, which is a generalization of deep kernel method by means of $C^*$-algebra. We can make use of the products in $C^*$-algebra to induce interactions among data components. We also show a representer theorem to guarantee the representation of solutions only with given data.
- We derive a generalization bound for deep RKHM. The dependency of the bound on the output dimension is milder than existing bounds by virtue of $C^*$-algebras. In addition, the Perron–Frobenius operators provide a connection of our bound with benign overfitting.
- We show connections of our study with existing studies such as CNNs and neural tangent kernel.

We emphasize that our theoretical analysis using $C^*$-algebra and Perron–Frobenius operators gives a new and powerful lens through which one can design and analyze kernel methods.

## 2 Preliminaries

### 2.1 $C^*$-algebra and reproducing kernel $C^*$-module

$C^*$-algebra, which is denoted by $\mathcal{A}$ in the following, is a Banach space equipped with a product and an involution satisfying the $C^*$ identity (condition 3 below).

**Definition 2.1** ($C^*$-**algebra**)  *A set $\mathcal{A}$ is called a $C^*$-algebra if it satisfies the following conditions:*

*1. $\mathcal{A}$ is an algebra over $\mathbb{C}$ and equipped with a bijection $(\cdot)^* : \mathcal{A} \to \mathcal{A}$ that satisfies the following conditions for $\alpha, \beta \in \mathbb{C}$ and $a, b \in \mathcal{A}$:*

- $(\alpha a + \beta b)^* = \overline{\alpha} a^* + \overline{\beta} b^*$,    • $(ab)^* = b^* a^*$,    • $(a^*)^* = a.$

2. $\mathcal{A}$ is a normed space endowed with $\|\cdot\|_{\mathcal{A}}$, and for $a, b \in \mathcal{A}$, $\|ab\|_{\mathcal{A}} \leq \|a\|_{\mathcal{A}}\|b\|_{\mathcal{A}}$ holds. In addition, $\mathcal{A}$ is complete with respect to $\|\cdot\|_{\mathcal{A}}$.

3. For $a \in \mathcal{A}$, the $C^*$ identity $\|a^*a\|_{\mathcal{A}} = \|a\|_{\mathcal{A}}^2$ holds.

**Example 2.2** *A typical example of $C^*$-algebras is the $C^*$-algebra of $d$ by $d$ circulant matrices. Another example is the $C^*$-algebra of $d$ by $d$ block matrices with $M$ blocks and their block sizes are $\mathbf{m} = (m_1, \ldots, m_M)$. We denote them by $Circ(d)$ and $Block(\mathbf{m}, d)$, respectively. See [27, 32] for more details about these examples.*

We now define RKHM. Let $\mathcal{X}$ be a non-empty set for data.

**Definition 2.3 ($\mathcal{A}$-valued positive definite kernel)** *An $\mathcal{A}$-valued map $k : \mathcal{X} \times \mathcal{X} \to \mathcal{A}$ is called a positive definite kernel if it satisfies the following conditions:*
- $k(x, y) = k(y, x)^*$ *for $x, y \in \mathcal{X}$,*
- $\sum_{i,j=1}^n c_i^* k(x_i, x_j) c_j$ *is positive semi-definite for $n \in \mathbb{N}$, $c_i \in \mathcal{A}$, $x_i \in \mathcal{X}$.*

Let $\phi : \mathcal{X} \to \mathcal{A}^{\mathcal{X}}$ be the *feature map* associated with $k$, defined as $\phi(x) = k(\cdot, x)$ for $x \in \mathcal{X}$ and let $\mathcal{M}_{k,0} = \{\sum_{i=1}^n \phi(x_i)c_i \mid n \in \mathbb{N},\ c_i \in \mathcal{A},\ x_i \in \mathcal{X}\ (i = 1, \ldots, n)\}$. We can define an $\mathcal{A}$-valued map $\langle\cdot, \cdot\rangle_{\mathcal{M}_k} : \mathcal{M}_{k,0} \times \mathcal{M}_{k,0} \to \mathcal{A}$ as

$$\Big\langle \sum_{i=1}^n \phi(x_i)c_i, \sum_{j=1}^l \phi(y_j)b_j \Big\rangle_{\mathcal{M}_k} := \sum_{i=1}^n \sum_{j=1}^l c_i^* k(x_i, y_j) b_j,$$

which enjoys the reproducing property $\langle \phi(x), f \rangle_{\mathcal{M}_k} = f(x)$ for $f \in \mathcal{M}_{k,0}$ and $x \in \mathcal{X}$. The *reproducing kernel Hilbert $\mathcal{A}$-module (RKHM)* $\mathcal{M}_k$ associated with $k$ is defined as the completion of $\mathcal{M}_{k,0}$. See, for example, the references [33, 34, 26] for more details about $C^*$-algebra and RKHM.

## 2.2 Perron–Frobenius operator on RKHM

We introduce Perron–Frobenius operator on RKHM [35]. Let $\mathcal{X}_1$ and $\mathcal{X}_2$ be nonempty sets and let $k_1$ and $k_2$ be $\mathcal{A}$-valued positive definite kernels. Let $\mathcal{M}_1$ and $\mathcal{M}_2$ be RKHMs associated with $k_1$ and $k_2$, respectively. Let $\phi_1$ and $\phi_2$ be the feature maps of $\mathcal{M}_1$ and $\mathcal{M}_2$, respectively. We begin with the standard notion of linearity in RKHMs.

**Definition 2.4 ($\mathcal{A}$-linear)** *A linear map $A : \mathcal{M}_1 \to \mathcal{M}_2$ is called $\mathcal{A}$-linear if for any $a \in \mathcal{A}$ and $w \in \mathcal{M}_1$, we have $A(wa) = (Aw)a$.*

**Definition 2.5 (Perron–Frobenius operator)** *Let $f : \mathcal{X}_1 \to \mathcal{X}_2$ be a map. The Perron–Frobenius operator with respect to $f$ is an $\mathcal{A}$-linear map $P_f : \mathcal{M}_1 \to \mathcal{M}_2$ satisfying*
$$P_f \phi_1(x) = \phi_2(f(x)).$$

Note that a Perron–Frobenius operator is not always well-defined since $ac = bc$ for $a, b \in \mathcal{A}$ and nonzero $c \in \mathcal{A}$ does not always mean $a = b$.

**Definition 2.6 ($\mathcal{A}$-linearly independent)** *The set $\{\phi_1(x) \mid x \in \mathcal{X}\}$ is called $\mathcal{A}$-linearly independent if it satisfies the following condition: For any $n \in \mathbb{N}$, $x_1, \ldots, x_n \in \mathcal{X}$, and $c_1, \ldots, c_n \in \mathcal{A}$, "$\sum_{i=1}^n \phi_1(x_i)c_i = 0$" is equivalent to "$c_i = 0$ for all $i = 1, \ldots, n$".*

**Lemma 2.7** *If $\{\phi_1(x) \mid x \in \mathcal{X}\}$ is $\mathcal{A}$-linearly independent, then $P_f$ is well-defined.*

The following lemma gives a sufficient condition for $\mathcal{A}$-linearly independence.

**Lemma 2.8** *Let $k_1 = \tilde{k}a$, i.e., $k$ is separable, for an invertible operator $a$ and a complex-valued kernel $\tilde{k}$. Assume $\{\tilde{\phi}(x) \mid x \in \mathcal{X}\}$ is linearly independent (e.g. $\tilde{k}$ is Gaussian or Laplacian), where $\tilde{\phi}$ is the feature map associated with $\tilde{k}$. Then, $\{\phi_1(x) \mid x \in \mathcal{X}\}$ is $\mathcal{A}$-linearly independent.*

Note that separable kernels are widely used in existing literature of vvRKHS (see e.g. [36]). Lemma 2.8 guarantees the validity of the analysis with Perron–Frobenius operators, at least in the separable case. This provides a condition for "good kernels" by means of the well-definedness of the Perron–Frobenius operator.

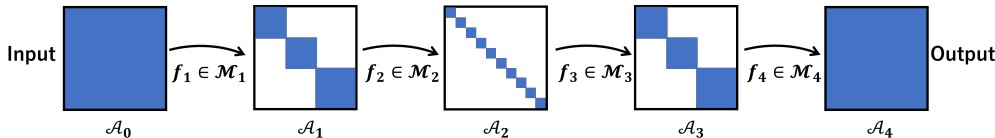

Figure 1: Overview of the proposed deep RKHM. The small blue squares represent matrix elements. In the case of the autoencoder (see Example 3.1), $f_1 \circ f_2$ is the encoder, and $f_3 \circ f_4$ is the decoder.

**Notation**  We denote the Euclidean inner product and norm by $\langle \cdot, \cdot \rangle$ and $\| \cdot \|$ without the subscript. For $a \in \mathcal{A}$, let $|a|_{\mathcal{A}}$ be the unique element in $\mathcal{A}$ satisfying $|a|_{\mathcal{A}}^2 = a^* a$. If $a$ is a matrix, $\|a\|_{\mathrm{HS}}$ is the Hilbert–Schmidt norm of $a$. The operator norm of a linear operator $A$ on an RKHM is denoted by $\|A\|_{\mathrm{op}}$. All the technical proofs are documented in the supplementary.

## 3  Deep RKHM

We now construct an $L$-layer deep RKHM. Let $\mathcal{A} = \mathbb{C}^{d \times d}$ be the $C^*$-algebra of $d$ by $d$ matrices. Let $\mathcal{A}_0, \ldots, \mathcal{A}_L$ be $C^*$-subalgebras of $\mathcal{A}$ and for $j = 1, \ldots, L$ and $k_j : \mathcal{A}_{j-1} \times \mathcal{A}_{j-1} \to \mathcal{A}_j$ be an $\mathcal{A}_j$-valued positive definite kernel for the $j$th layer. For $j = 1, \ldots, L$, we denote by $\mathcal{M}_j$ the RKHM over $\mathcal{A}_j$ associated with $k_j$. In addition, we denote by $\tilde{\mathcal{M}}_j$ the RKHM over $\mathcal{A}$ associated with $k_j$. Note that $\mathcal{M}_j$ is a subspace of $\tilde{\mathcal{M}}_j$ and for $u, v \in \mathcal{M}_j$, we have $\langle u, v \rangle_{\tilde{\mathcal{M}}_j} = \langle u, v \rangle_{\mathcal{M}_j}$. We set the function space corresponding to each layer as $\mathcal{F}_L = \{ f \in \mathcal{M}_L \mid f(x) \in \mathbb{R}^{d \times d} \text{ for any } x \in \mathcal{A}_{L-1}, \|f\|_{\mathcal{M}_L} \leq B_L \}$ and $\mathcal{F}_j = \{ f \in \mathcal{M}_j \mid f(x) \in \mathbb{R}^{d \times d} \text{ for any } x \in \mathcal{A}_{j-1}, \|P_f\|_{\mathrm{op}} \leq B_j \}$ for $j = 1, \ldots, L-1$. Here, for $f \in \mathcal{M}_j$ with $j = 1, \ldots, L-1$, $P_f$ is the Perron–Frobenius operator with respect to $f$ from $\tilde{\mathcal{M}}_j$ to $\tilde{\mathcal{M}}_{j+1}$. We assume the well-definedness of these operators. Then, we set the class of deep RKHM as

$$\mathcal{F}_L^{\mathrm{deep}} = \{ f_L \circ \cdots \circ f_1 \mid f_j \in \mathcal{F}_j \ (j = 1, \ldots, L) \}.$$

Figure 1 schematically shows the structure of the deep RKHM.

**Example 3.1**  *We can set $\mathcal{A}_j = Block((m_{1,j}, \ldots, m_{M_j,j}), d)$, the $C^*$-algebra of block diagonal matrices. By setting $M_1 \leq \cdots \leq M_l$ and $M_l \geq \cdots \geq M_L$ for $l < L$, the number of nonzero elements in $\mathcal{A}_j$ decreases during $1 \leq j \leq l$ and increases during $l \leq j \leq L$. This construction is regarded as an autoencoder, where the $1 \sim l$th layer corresponds to the encoder and the $l+1 \sim L$th layer corresponds to the decoder.*

**Advantage over existing deep kernel methods**  Note that deep kernel methods with RKHSs and vvRKHSs have been proposed [1, 4, 2]. Autoencoders using RKHSs and vvRKHSs were also proposed [3, 5]. We have at least three advantages of deep RKHM over deep vvRKHSs or RKHSs: 1) useful structures of matrices stated in Remark 5.2, 2) availability of the operator norm in the generalization bound stated in Section 4, 3) connection with CNNs stated in Subsection 6.1.

## 4  Generalization Bound

We derive a generalization error for deep RKHM. To derive the bound, we bound the Rademacher complexity, which is one of the typical methods on deriving generalization bounds [11]. Let $\Omega$ be a probability space equipped with a probability measure $P$. We denote the integral $\int_{\Omega} s(\omega) \mathrm{d}P(\omega)$ of a measurable function $s$ on $\Omega$ by $\mathrm{E}[s]$. Let $x_1, \ldots, x_n$ and $y_1, \ldots, y_n$ be input and output samples from the distributions of $\mathcal{A}_0$ and $\mathcal{A}_L$-valued random variables $x$ and $y$, respectively. Let $\sigma_{i,j}$ $(i = 1, \ldots, d, \ j = 1, \ldots, n)$ be i.i.d. Rademacher variables. Let $\sigma_j = (\sigma_{1,j}, \ldots, \sigma_{d,j})$. For an $\mathbb{R}^d$-valued function class $\mathcal{F}$ and $\mathbf{x} = (x_1, \ldots, x_n)$, the empirical Rademacher complexity $\hat{R}_n(\mathbf{x}, \mathcal{F})$ is defined by $\hat{R}_n(\mathbf{x}, \mathcal{F}) := \mathrm{E}[\sup_{f \in \mathcal{F}} \sum_{i=1}^{n} \langle \sigma_i, f(x_i) \rangle]/n$.

### 4.1  Bound for shallow RKHMs

We use the operator norm to derive a bound, whose dependency on output dimension is milder than existing bounds. Availability of the operator norm is one of the advantages of considering

$C^*$-algebras (RKHMs) instead of vectors (vvRKHSs). Note that although we can also use the Hilbert–Schmidt norm for matrices, it tends to be large as the dimension $d$ becomes large. On the other hand, the operator norm is defined independently of the dimension $d$. Indeed, the Hilbert–Schmidt norm of a matrix $a \in \mathbb{C}^{d \times d}$ is calculated as $(\sum_{i=1}^{d} s_i^2)^{1/2}$, where $s_i$ is the $i$th singular value of $a$. The operator norm of $a$ is the largest singular value of $a$.

To see the derivation of the bound, we first focus on the case of $L = 1$, i.e., the network is shallow. Let $E > 0$. For a space $\mathcal{F}$ of $\mathcal{A}$-valued functions on $\mathcal{A}_0$, let $\mathcal{G}(\mathcal{F}) = \{(x, y) \mapsto f(x) - y \mid f \in \mathcal{F}, \|y\|_{\mathcal{A}} \le E\}$. The following theorem shows a bound for RKHMs with the operator norm.

**Theorem 4.1** *Assume there exists $D > 0$ such that $\|k_1(x, x)\|_{\mathcal{A}} \le D$ for any $x \in \mathcal{A}_0$. Let $\tilde{K} = 4\sqrt{2}(\sqrt{D}B_1 + E)B_1$ and $\tilde{M} = 6(\sqrt{D}B_1 + E)^2$. Let $\delta \in (0, 1)$. Then, for any $g \in \mathcal{G}(\mathcal{F}_1)$, where $\mathcal{F}_1$ is defined in Section 3, with probability at least $1 - \delta$, we have*

$$\|\mathrm{E}[|g(x, y)|_{\mathcal{A}}^2]\|_{\mathcal{A}} \le \left\| \frac{1}{n} \sum_{i=1}^{n} |g(x_i, y_i)|_{\mathcal{A}}^2 \right\|_{\mathcal{A}} + \frac{\tilde{K}}{n} \left( \sum_{i=1}^{n} \mathrm{tr}\, k_1(x_i, x_i) \right)^{1/2} + \tilde{M} \sqrt{\frac{\log(2/\delta)}{2n}}.$$

Theorem 4.1 is derived by the following lemmas. We first fix a vector $p \in \mathbb{R}^d$ and consider the operator-valued loss function acting on $p$. We first show a relation between the generalization error and the Rademacher complexity of vector-valued functions. Then, we bound the Rademacher complexity. Since the bound does not depend on $p$, we can finally remove the dependency on $p$.

**Lemma 4.2** *Let $\mathcal{F}$ be a function class of $\mathbb{R}^{d \times d}$-valued functions on $\mathcal{A}_0$ bounded by $C$ (i.e., $\|f(x)\|_{\mathcal{A}} \le C$ for any $x \in \mathcal{A}_0$). Let $\tilde{\mathcal{G}}(\mathcal{F}, p) = \{(x, y) \mapsto \|(f(x) - y)p\|^2 \mid f \in \mathcal{F}, \|y\|_{\mathcal{A}} \le E\}$ and $M = 2(C + E)^2$. Let $p \in \mathbb{R}^d$ satisfy $\|p\| = 1$ and let $\delta \in (0, 1)$. Then, for any $g \in \tilde{\mathcal{G}}(\mathcal{F}, p)$, with probability at least $1 - \delta$, we have*

$$\|\mathrm{E}[|g(x, y)|_{\mathcal{A}}^2]^{1/2}p\|^2 \le \left\| \frac{1}{n} \sum_{i=1}^{n} |g(x_i, y_i)|_{\mathcal{A}}^2 \right\|_{\mathcal{A}} + 2\hat{R}_n(\mathbf{x}, \tilde{\mathcal{G}}(\mathcal{F}, p)) + 3M \sqrt{\frac{\log(2/\delta)}{2n}}.$$

**Lemma 4.3** *With the same notations in Lemma 4.2, let $K = 2\sqrt{2}(C + E)$. Then, we have $\hat{R}_n(\mathbf{x}, \mathcal{G}(\mathcal{F}, p)) \le K\hat{R}_n(\mathbf{x}, \mathcal{F}p)$, where $\mathcal{F}p = \{x \mapsto f(x)p \mid f \in \mathcal{F}\}$.*

**Lemma 4.4** *Let $p \in \mathbb{R}^d$ satisfy $\|p\| = 1$. For $\mathcal{F}_1$ defined in Section 3, we have*

$$\hat{R}_n(\mathbf{x}, \mathcal{F}_1 p) \le \frac{B_1}{n} \left( \sum_{i=1}^{n} \mathrm{tr}(k(x_i, x_i)) \right)^{1/2}.$$

## 4.2 Bound for deep RKHMs

We now generalize Theorem 4.1 to the deep setting ($L \ge 2$) using the Perron–Frobenius operators.

**Theorem 4.5** *Assume there exists $D > 0$ such that $\|k_L(x, x)\|_{\mathcal{A}} \le D$ for any $x \in \mathcal{A}_0$. Let $\tilde{K} = 4\sqrt{2}(\sqrt{D}B_L + E)B_1 \cdots B_L$ and $\tilde{M} = 6(\sqrt{D}B_L + E)^2$. Let $\delta \in (0, 1)$. Then, for any $g \in \mathcal{G}(\mathcal{F}_L^{\mathrm{deep}})$, with probability at least $1 - \delta$, we have*

$$\|\mathrm{E}[|g(x, y)|_{\mathcal{A}}^2]\|_{\mathcal{A}} \le \left\| \frac{1}{n} \sum_{i=1}^{n} |g(x_i, y_i)|_{\mathcal{A}}^2 \right\|_{\mathcal{A}} + \frac{\tilde{K}}{n} \left( \sum_{i=1}^{n} \mathrm{tr}\, k_1(x_i, x_i) \right)^{1/2} + \tilde{M} \sqrt{\frac{\log(2/\delta)}{2n}}.$$

We use the following proposition and Lemmas 4.2 and 4.3 to show Theorem 4.5. The key idea of the proof is that by the reproducing property and the definition of the Perron–Frobenius operator, we get $f_L \circ \cdots \circ f_1(x) = \langle \phi_L(f_{L-1} \circ \cdots \circ f_1(x)), f_L \rangle_{\tilde{\mathcal{M}}_L} = \langle P_{f_{L-1}} \cdots P_{f_1} \phi(x), f_L \rangle_{\tilde{\mathcal{M}}_L}$.

**Proposition 4.6** *Let $p \in \mathbb{R}^d$ satisfy $\|p\| = 1$. Then, we have*

$$\hat{R}_n(\mathbf{x}, \mathcal{F}_L^{\mathrm{deep}} p) \le \frac{1}{n} \sup_{(f_j \in \mathcal{F}_j)_j} \|P_{f_{L-1}} \cdots P_{f_1}|_{\tilde{\mathcal{V}}(\mathbf{x})}\|_{\mathrm{op}} \|f_L\|_{\mathcal{M}_L} \left( \sum_{i=1}^{n} \mathrm{tr}(k_1(x_i, x_i)) \right)^{1/2}.$$

*Here, $\tilde{\mathcal{V}}(\mathbf{x})$ is the submodule of $\tilde{\mathcal{M}}_1$ generated by $\phi_1(x_1), \ldots \phi_1(x_n)$.*

**Corollary 4.7** *Let $p \in \mathbb{R}^d$ satisfy $\|p\| = 1$. Then, we have*

$$\hat{R}_n(\mathbf{x}, \mathcal{F}_L^{\text{deep}}p) \leq \frac{1}{n} B_1 \cdots B_L \left( \sum_{i=1}^n \text{tr}(k_1(x_i, x_i)) \right)^{1/2}.$$

**Comparison to deep vvRKHS**  We can also regard $\mathbb{C}^{d \times d}$ as the Hilbert space equipped with the Hilbert–Schmidt inner product, i.e., we can flatten matrices and get $d^2$-dimensional Hilbert space. In this case, the corresponding operator-valued kernel is the multiplication operator of $k(x, y)$, which we denote by $M_{k(x,y)}$. Then, we can apply existing results for vvRKHSs [5, 12], which involve the term $(\sum_{i=1}^n \text{tr}(M_{k(x_i, x_i)}))^{1/2}$. It is calculated as

$$\sum_{i=1}^n \text{tr}(M_{k(x_i, x_i)}) = \sum_{i=1}^n \sum_{j,l=1}^d \langle e_{jl}, k(x_i, x_i)e_{jl} \rangle_{\text{HS}} = \sum_{i=1}^n \sum_{j,l=1}^d k(x_i, x_i)_{l,l} = d \sum_{i=1}^n \text{tr}\, k(x_i, x_i).$$

Thus, using the existing approaches, we have the factor $(d \sum_{i=1}^n \text{tr}\, k(x_i, x_i))^{1/2}$. On the other hand, we have the smaller factor $(\sum_{i=1}^n \text{tr}\, k(x_i, x_i))^{1/2}$ in Theorems 4.1 and 4.5. Using the operator norm, we can reduce the dependency on the dimension $d$.

## 5   Learning Deep RKHMs

We focus on the practical learning problem. To learn deep RKHMs, we consider the following minimization problem based on the generalization bound derived in Section 4:

$$\min_{(f_j \in \mathcal{M}_j)_j} \left\| \frac{1}{n} \sum_{i=1}^n |f_L \circ \cdots \circ f_1(x_i) - y_i|_{\mathcal{A}}^2 \right\|_{\mathcal{A}} + \lambda_1 \|P_{f_{L-1}} \cdots P_{f_1}|_{\tilde{\mathcal{V}}(\mathbf{x})}\|_{\text{op}} + \lambda_2 \|f_L\|_{\mathcal{M}_L}. \quad (1)$$

The second term regarding the Perron–Frobenius operators comes from the bound in Proposition 4.6. We try to reduce the generalization error by reducing the magnitude of the norm of the Perron–Frobenius operators.

### 5.1   Representer theorem

We first show a representer theorem to guarantee that a solution of the minimization problem (1) is represented only with given samples.

**Proposition 5.1** *Let $h : \mathcal{A}^n \times \mathcal{A}^n \to \mathbb{R}_+$ be an error function, let $g_1$ be an $\mathbb{R}_+$-valued function on the space of bounded linear operators on $\tilde{\mathcal{M}}_1$, and let $g_2 : \mathbb{R}_+ \to \mathbb{R}_+$ satisfy $g_2(a) \leq g_2(b)$ for $a \leq b$. Assume the following minimization problem has a solution:*

$$\min_{(f_j \in \mathcal{M}_j)_j} h(f_L \circ \cdots \circ f_1(x_1), \ldots, f_L \circ \cdots \circ f_1(x_n)) + g_1(P_{f_{L-1}} \cdots P_{f_L}|_{\tilde{\mathcal{V}}(\mathbf{x})}) + g_2(\|f_L\|_{\mathcal{M}_L}).$$

*Then, there exists a solution admitting a representation of the form $f_j = \sum_{i=1}^n \phi_j(x_i^{j-1})c_{i,j}$ for some $c_{1,j}, \ldots, c_{n,j} \in \mathcal{A}$ and for $j = 1, \ldots, L$. Here, $x_i^j = f_j \circ \cdots \circ f_1(x_i)$ for $j = 1, \ldots, L$ and $x_i^0 = x_i$.*

**Remark 5.2** *An advantage of deep RKHM compared to deep vvRKHS is that we can make use of the structure of matrices. For example, the product of two diagonal matrices is calculated by the element-wise product of diagonal elements. Thus, when $\mathcal{A}_1 = \cdots = \mathcal{A}_L = Block((1, \ldots, 1), d)$, interactions among elements in an input are induced only by the kernels, not by the product $k_j(x, x_i^{j-1}) \cdot c_{i,j}$. That is, the form of interactions does not depend on the learning parameter $c_{i,j}$. On the other hand, if we set $\mathcal{A}_j = Block(\mathbf{m}_j, d)$ with $\mathbf{m}_j = (m_{1,j}, \ldots, m_{M_j,j}) \neq (1, \ldots, 1)$, then at the jth layer, we get interactions among elements in the same block through the product $k_j(x, x_i^{j-1}) \cdot c_{i,j}$. In this case, the form of interactions is learned through $c_{i,j}$. For example, the part $M_1 \leq \cdots \leq M_l$ (the encoder) in Example 3.1 tries to gradually reduce the dependency among elements in the output of each layer and describe the input with a small number of variables. The part $M_l \geq \cdots \geq M_L$ (the decoder) tries to increase the dependency.*

## 5.2 Computing the norm of the Perron–Frobenius operator

We discuss the practical computation and the role of the factor $\|P_{f_{L-1}} \cdots P_{f_1}|_{\tilde{\mathcal{V}}(\mathbf{x})}\|_{\mathrm{op}}$ in Eq. (1). Let $G_j \in \mathcal{A}^{n \times n}$ be the Gram matrix whose $(i, l)$-entry is $k_j(x_i^{j-1}, x_l^{j-1}) \in \mathcal{A}$.

**Proposition 5.3** *For $j = 1, L$, let $[\phi_j(x_1^{j-1}), \ldots, \phi_j(x_n^{j-1})]R_j = Q_j$ be the QR decomposition of $[\phi_j(x_1^{j-1}), \ldots, \phi_j(x_n^{j-1})]$. Then, we have $\|P_{f_{L-1}} \cdots P_{f_1}|_{\tilde{\mathcal{V}}(\mathbf{x})}\|_{\mathrm{op}} = \|R_L^* G_L R_1\|_{\mathrm{op}}$.*

The computational cost of $\|R_L^* G_L R_1\|_{\mathrm{op}}$ is expensive if $n$ is large since computing $R_1$ and $R_L$ is expensive. Thus, we consider upper bounding $\|R_L^* G_L R_1\|_{\mathrm{op}}$ by a computationally efficient value.

**Proposition 5.4** *Assume $G_L$ is invertible. Then, we have*

$$\|P_{f_{L-1}} \cdots P_{f_1}|_{\tilde{\mathcal{V}}(\mathbf{x})}\|_{\mathrm{op}} \leq \|G_L^{-1}\|_{\mathrm{op}}^{1/2} \|G_L\|_{\mathrm{op}} \|G_1^{-1}\|_{\mathrm{op}}^{1/2}. \tag{2}$$

Since $G_1$ is independent of $f_1, \ldots, f_L$, to make the value $\|P_{f_{L-1}} \cdots P_{f_1}|_{\tilde{\mathcal{V}}(\mathbf{x})}\|_{\mathrm{op}}$ small, we try to make the norm of $G_L$ and $G_L^{-1}$ small according to Proposition 5.4. For example, instead of the second term regarding the Perron–Frobenius operators in Eq. (1), we can consider the following term with $\eta > 0$:

$$\lambda_1(\|(\eta I + G_L)^{-1}\|_{\mathrm{op}} + \|G_L\|_{\mathrm{op}}).$$

Note that the term depends on the training samples $x_1, \ldots, x_n$. The situation is different from the third term in Eq. (1) since the third term does not depend on the training samples before applying the representer theorem. If we try to minimize a value depending on the training samples, the model seems to be more specific for the training samples, and it may cause overfitting. Thus, the connection between the minimization of the second term in Eq. (1) and generalization cannot be explained by the classical argument about generalization and regularization. However, as we will see in Subsection 6.2, it is related to benign overfitting and has a good effect on generalization.

**Remark 5.5** *The inequality (2) implies that as $\{\phi_L(x_1^{L-1}), \ldots, \phi_L(x_n^{L-1})\}$ becomes nearly linearly dependent, the Rademacher complexity becomes large. By the term $\|(\eta I + G_L)^{-1}\|$ in Eq. (1), the function $f_{L-1} \circ \cdots \circ f_1$ is learned so that it separates $x_1, \ldots, x_n$ well.*

**Remark 5.6** *To evaluate $f_1 \circ \cdots \circ f_L(x_i)$ for $i = 1, \ldots, n$, we have to construct a Gram matrix $G_j \in \mathcal{A}_j^{n \times n}$, and compute the product of the Gram matrix and a vector $c_j \in \mathcal{A}_j^n$ for $j = 1, \ldots, L$. The computational cost of the construction of the Gram matrices does not depend on $d$. The cost of computing $G_j c_j$ is $O(n^2 \tilde{d}_j)$, where $\tilde{d}_j$ is the number of nonzero elements in the matrices in $\mathcal{A}_j$. Note that if $\mathcal{A}_j$ is the $C^*$-algebra of block diagonal matrices, then we have $\tilde{d}_j \ll d^2$. Regarding the cost with respect to $n$, if the positive definite kernel is separable, we can use the random Fourier features [37] to replace the factor $n^2$ with $mn$ for a small integer $m \ll n$.*

## 6 Connection and Comparison with Existing Studies

The proposed deep RKHM is deeply related to existing studies by virtue of $C^*$-algebra and the Perron–Frobanius operators. We discuss the connection below.

### 6.1 Connection with CNN

The proposed deep RKHM has a duality with CNNs. Let $\mathcal{A}_0 = \cdots = \mathcal{A}_L = Circ(d)$, the $C^*$-algebra of $d$ by $d$ circulant matrices. For $j = 1, \ldots, L$, let $a_j \in \mathcal{A}_j$. Let $k_j$ be an $\mathcal{A}_j$-valued positive definite kernel defined as $k_j(x, y) = \tilde{k}_j(a_j x, a_j y)$, where $\tilde{k}_j$ is an $\mathcal{A}_j$-valued function. The $\mathcal{A}_j$-valued positive definite kernel makes the output of each layer become a circulant matrix, which enables us to apply the convolution as the product of the output and a parameter. Then, $f_j$ is represented as $f_j(x) = \sum_{i=1}^n \tilde{k}_j(a_j x, a_j x_i^{j-1}) c_{i,j}$ for some $c_{i,j} \in \mathcal{A}_j$. Thus, at the $j$th layer, the input $x$ is multiplied by $a_j$ and is transformed nonlinearly by $\sigma_j(x) = \sum_{i=1}^n k_j(x, a_j x_i^{j-1}) c_{i,j}$. Since the product of two circulant matrices corresponds to the convolution, $a_j$ corresponds to a filter. In

addition, $\sigma_j$ corresponds to the activation function at the $j$th layer. Thus, the deep RKHM with the above setting corresponds to a CNN. The difference between the deep RKHM and the CNN is the parameters that we learn. Whereas for the deep RKHM, we learn the coefficients $c_{1,j}, \ldots, c_{n,j}$, for the CNN, we learn the parameter $a_j$. In other words, whereas for the deep RKHM, we learn the activation function $\sigma_j$, for the CNN, we learn the filter $a_j$. It seems reasonable to interpret this difference as a consequence of solving the problem in the primal or in the dual. In the primal, the number of the learning parameters depends on the dimension of the data (or the filter for convolution), while in the dual, it depends on the size of the data.

**Remark 6.1** *The connection between CNNs and shallow RKHMs has already been studied [27]. However, the existing study does not provide the connection between the filter and activation function. The above investigation shows a more clear layer-wise connection of deep RKHMs with CNNs.*

## 6.2 Connection with benign overfitting

Benign overfitting is a phenomenon that the model fits any amount of data yet generalizes well [20, 21]. For kernel regression, Mallinar et al. [22] showed that if the eigenvalues of the integral operator associated with the kernel function over the data distribution decay slower than any powerlaw decay, than the model exhibits benign overfitting. The Gram matrix is obtained by replacing the integral with the sum over the finite samples. The inequality (2) suggests that the generalization error becomes smaller as the smallest and the largest eigenvalues of the Gram matrix get closer, which means the eigenvalue decay is slower. Combining the observation in Remark 5.5, we can interpret that as the right-hand side of the inequality (2) becomes smaller, the outputs of noisy training data at the $L-1$th layer tend to be more separated from the other outputs. In other words, for the random variable $x$ following the data distribution, $f_{L-1} \circ \cdots \circ f_1$ is learned so that the distribution of $f_{L-1} \circ \cdots \circ f_1(x)$ generates the integral operator with more separated eigenvalues, which appreciates benign overfitting. We will also observe this phenomenon numerically in Section 7 and Appendix C.3.2. Since the generalization bound for deep vvRKHSs [5] is described by the Lipschitz constants of the feature maps and the norm of $f_j$ for $j = 1, \ldots, L$, this type of theoretical interpretation regarding benign overfitting is not available for the existing bound for deep vvRKHSs.

**Remark 6.2** *The above arguments about benign overfitting are valid only for deep RKHMs, i.e., the case of $L \geq 2$. If $L = 1$ (shallow RKHM), then the Gram matrix $G_L = [k(x_i, x_j)]_{i,j}$ is fixed and determined only by the training data and the kernel. On the other hand, if $L \geq 2$ (deep RKHM), then $G_L = [k_L(f_{L-1} \circ \cdots \circ f_1(x_i), f_{L-1} \circ \cdots \circ f_1(x_j))]_{i,j}$ depends also on $f_1, \ldots, f_{L-1}$. As a result, by adding the term using $G_L$ to the loss function, we can learn proper $f_1, \ldots, f_{L-1}$ so that they make the right-hand side of Eq. (2) small, and the whole network overfits benignly. As $L$ becomes large, the function $f_{L-1} \circ \cdots \circ f_1$ changes more flexibly to attain a smaller value of the term. This is an advantage of considering a large $L$.*

## 6.3 Connection with neural tangent kernel

Neural tangent kernel has been investigated to understand neural networks using the theory of kernel methods [6, 7]. Generalizing neural networks to $C^*$-algebra, which is called the $C^*$-algebra network, is also investigated [32, 38]. We define a neural tangent kernel for the $C^*$-algebra network and develop a theory for combining neural networks and deep RKHMs as an analogy of the existing studies. Consider the $C^*$-algebra network $f : \mathcal{A}^{N_0} \to \mathcal{A}$ over $\mathcal{A}$ with $\mathcal{A}$-valued weight matrices $W_j \in \mathcal{A}^{N_j \times N_{j-1}}$ and element-wise activation functions $\sigma_j$: $f(x) = W_L \sigma_{L-1}(W_{L-1} \cdots \sigma_1(W_1 x) \cdots)$. The $(i, j)$-entry of $f(x)$ is $f_i(\mathbf{x}_j) = \mathbf{W}_{L,i} \sigma_{L-1}(W_{L-1} \cdots \sigma_1(W_1 \mathbf{x}_j) \cdots)$, where $\mathbf{x}_j$ is the $j$th column of $x$ regarded as $x \in \mathbb{C}^{dN_0 \times d}$ and $\mathbf{W}_{L,i}$ is the $i$th row of $W_L$ regarded as $W_L \in \mathbb{C}^{d \times dN_{L-1}}$. Thus, the $(i, j)$-entry of $f(x)$ corresponds to the output of the network $f_i(\mathbf{x}_j)$. We can consider the neural tangent kernel for each $f_i$. Chen and Xu [7] showed that the RKHS associated with the neural tangent kernel restricted to the sphere is the same set as that associated with the Laplacian kernel. Therefore, the $i$th row of $f(x)$ is described by the shallow RKHS associated with the neural tangent kernel $k_i^{\mathrm{NT}}$. Let $k^{\mathrm{NN}}$ be the $\mathcal{A}$-valued positive definite kernel whose $(i, j)$-entry is $k_i^{\mathrm{NT}}$ for $i = j$ and 0 for $i \neq j$. Then, for any function $g \in \mathcal{M}_{k^{\mathrm{NN}}}$, the elements in the $i$th row of $g$ are in the RKHS associated with $k_i^{\mathrm{NT}}$, i.e., associated with the Laplacian kernel. Thus, $f$ is described by this shallow RKHM.

**Remark 6.3** *We can combine the deep RKHM and existing neural networks by replacing some $f_j$ in our model with an existing neural network. The above observation enables us to apply our results in Section 4 to the combined network.*

### 6.4 Comparison to bounds for classical neural networks

Existing bounds for classical neural networks typically depend on the product or sum of matrix $(p, q)$ norm of all the weight matrices $W_j$ [14, 15, 16, 17]. Typical bound is $O(\sqrt{1/n} \prod_{j=1}^{L} \|W_j\|_{\mathrm{HS}})$. Note that the Hilbert–Schmidt norm is the matrix $(2, 2)$ norm. Unlike the operator norm, the matrix $(p, q)$ norm tends to be large as the width of the layers becomes large. On the other hand, the dependency of our bound on the width of the layer is not affected by the number of layers in the case where the kernels are separable. Indeed, assume we set $k_1 = \tilde{k}_1 a_1$ and $k_L = \tilde{k}_L a_L$ for some complex-valued kernels $\tilde{k}_1$ and $\tilde{k}_2$, $a_1 \in \mathcal{A}_1$, and $a_L \in \mathcal{A}_L$. Then by Proposition 5.3, the factor $\alpha(L) := \|P_{f_{L-1}} \cdots P_{f_1}|_{\tilde{\mathcal{V}}(\mathbf{x})}\|_{\mathrm{op}}$ is written as $\|\tilde{R}_L^* \tilde{G}_L \tilde{R}_1 \otimes a_L^2 a_1\|_{\mathrm{op}} = \|\tilde{R}_L^* \tilde{G}_L \tilde{R}_1\|_{\mathrm{op}} \|a_L^2 a_1\|_{\mathcal{A}}$ for some $\tilde{R}_L, \tilde{G}_L, \tilde{R}_1 \in \mathbb{C}^{n \times n}$. Thus, it is independent of $d$. The only part depending on $d$ is $\operatorname{tr} k_1(x_i, x_i)$, which results in the bound $O(\alpha(L)\sqrt{d/n})$. Note that the width of the $j$th layer corresponds to the number of nonzero elements in a matrix in $\mathcal{A}_j$. We also discuss in Appendix B the connection of our bound with the bound by Koopman operators, the adjoints of the Perron–Frobenius operators [18].

## 7 Numerical Results

We numerically confirm our theory and the validity of the proposed deep RKHM.

**Comparison to vvRKHS** We compared the generalization property of the deep RKHM to the deep vvRKHS with the same positive definite kernel. For $d = 10$ and $n = 10$, we set $x_i = (az_i)^2 + \epsilon_i$ as input samples, where $a \in \mathbb{R}^{100 \times 10}$ and $z_i \in \mathbb{R}^{10}$ are randomly generated by $\mathcal{N}(0, 0.1)$, the normal distribution of mean 0 and standard deviation 0.1, $(\cdot)^2$ is the elementwise product, and $\epsilon_i$ is the random noise drawn from $\mathcal{N}(0, 1e-3)$. We reshaped $x_i$ to a 10 by 10 matrix. We set $L = 3$ and $k_j = \tilde{k}I$ for $j = 1, 2, 3$, where $\tilde{k}$ is the Laplacian kernel. For RKHMs, we set $\mathcal{A}_1 = Block((1, \ldots, 1), d)$, $\mathcal{A}_2 = Block((2, \ldots, 2), d)$, and $\mathcal{A}_3 = \mathbb{C}^{d \times d}$. This is the autoencoder mentioned in Example 3.1. For vvRKHSs, we set the corresponding Hilbert spaces with the Hilbert–Schmidt inner product. We set the loss function as $1/n \| \sum_{i=1}^{n} |f(x_i) - x_i|_{\mathcal{A}}^2 \|_{\mathcal{A}}$ for the deep RKHM and as $1/n \sum_{i=1}^{n} \|f(x_i) - x_i\|_{\mathrm{HS}}^2$ for the deep vvRKHS. Here, $f = f_3 \circ f_2 \circ f_1$. We did not add any terms to the loss function to see how the loss function with the operator norm affects the generalization performance. We computed the same value $\|\mathrm{E}[|f(x) - x|_{\mathcal{A}}^2]\|_{\mathcal{A}} - 1/n \| \sum_{i=1}^{n} |f(x_i) - x_i|_{\mathcal{A}}^2 \|_{\mathcal{A}}$ for both RKHM and vvRKHS. Figure 2 (a) shows the results. We can see that the deep RKHM generalizes better than the deep vvRKHS, only with the loss function and without any additional terms.

**Observation about benign overfitting** We analyzed the overfitting numerically. For $d = 10$ and $n = 1000$, we randomly sampled $d$ by $d$ diagonal matrices $x_1, \ldots, x_n \in \mathcal{A}_0$ from the normal distribution $\mathcal{N}(0, 0.1)$. We set $y_i = x_i^2 + \epsilon_i$ for $i = 1, \ldots, n$, where $\epsilon_i$ is a noise drawn from the normal distribution $\mathcal{N}(0, 0.001)$. The magnitude of the noise is 10% of $x_i^2$. In addition, we set $L = 2$, $\mathcal{A}_1 = \mathbb{C}^{d \times d}$, $\mathcal{A}_2 = Block((1, \ldots, 1), d)$, and $k_j$ as the same kernel as the above experiment. The additional term to the loss function is set as $\lambda_1(\|(\eta I + G_L)^{-1}\|_{\mathrm{op}} + \|G_L\|_{\mathrm{op}}) + \lambda_2 \|f_L\|_{\mathcal{M}_L}^2$, where $\eta = 0.01$ and $\lambda_2 = 0.01$ according to Subsection 5.2. We computed the generalization error for the cases of $\lambda_1 = 0$ and $\lambda_1 = 10^2$. Figure 2 (b) shows the result. We can see that the generalization error saturates without the additional term motivated by the Perron–Frobenius operator. On the other hand, with the additional term, the generalization error becomes small, which is the effect of benign overfitting.

**Comparison to CNN** We compared the deep RKHM to a CNN on the classification task with MNIST [39]. We set $d = 28$ and $n = 20$. We constructed a deep RKHM combined with a neural network with 2 dense layers. For the deep RKHM, we set $L = 2$, $\mathcal{A}_0 = \mathbb{C}^{d \times d}$, $\mathcal{A}_1 = Block((7, 7, 7, 7), d)$, and $\mathcal{A}_2 = Block((4, \ldots, 4), d)$. Then, two dense layers are added. See Subsection 6.3 about combining the deep RKHM with neural networks. Regarding the additional term to the loss function, we set the same term with the previous experiment with $\lambda_2 = 0.001$ and

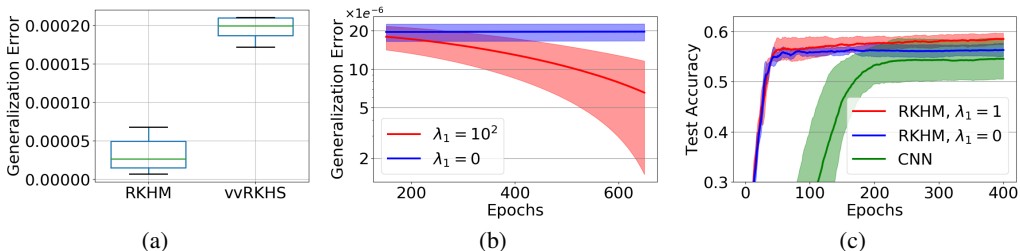

<p style="text-align:center">(a)           (b)           (c)</p>

Figure 2: (a) The box plot of the generalization error of the deep RKHM and vvRKHS at the point that the training error reaches 0.05. (b) Behavior of the generalization error during the learning process with and without the additional term regarding the Perron–Frobenius operators. (c) Test accuracy of the classification task with MNIST for a deep RKHM and a CNN.

set $\lambda_1 = 1$ or $\lambda_1 = 0$. To compare the deep RKHM to CNNs, we also constructed a network by replacing the deep RKHM with a CNN. The CNN is composed of 2 layers with $7 \times 7$ and $4 \times 4$ filters. Figure 2 (c) shows the test accuracy of these networks. We can see that the deep RKHM outperforms the CNN. In addition, we can see that the test accuracy is higher if we add the term regarding the Perron–Frobenius operators. We discuss the memory consumption and the computational cost of each of the deep RKHM and the CNN in Appendix C.3, and we empirically show that the deep RKHM outperforms the CNN that has the same size of learning parameters as the deep RKHM. We also show additional results about benign overfitting in Appendix C.3.

## 8    Conclusion and Limitations

In this paper, we proposed deep RKHM and analyzed it through $C^*$-algebra and the Perron–Frobenius operators. We derived a generalization bound, whose dependency on the output dimension is alleviated by the operator norm, and which is related to benign overfitting. We showed a representer theorem about the proposed deep RKHM, and connections with existing studies such as CNNs and neural tangent kernel. Our theoretical analysis shows that $C^*$-algebra and Perron–Frobenius operators are effective tools for analyzing deep kernel methods. The main contributions of this paper are our theoretical results with $C^*$-algebra and the Perron–Frobenius operators. More practical investigations are required for further progress. For example, although we numerically showed the validity of our method for the case where the number of samples is limited (the last experiment in Section 7), more experimental results for the case where the number of samples is large are useful for further analysis. Also, although we can apply random Fourier features (Remark 5.6) to reduce the computational costs, studying more efficient methods specific to deep RKHM remains to be investigated in future work. As for the theoretical topic, we assumed the well-definedness of the Perron–Frobenius operators. Though separable kernels with invertible matrices, which are typical examples of kernels, satisfy the assumption, generalization of our analysis to other kernels should also be studied in future work.

## Acknowledgements

Hachem Kadri is partially supported by grant ANR-19-CE23-0011 from the French National Research Agency. Masahiro Ikeda is partially supported by grant JPMJCR1913 from JST CREST.

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

# A  Proofs

We provide the proofs of the theorems, propositions, and lemmas in the main paper.

**Lemma 2.7**  *If $\{\phi_1(x) \mid x \in \mathcal{X}\}$ is $\mathcal{A}$-linearly independent, then $P_f$ is well-defined.*

**Proof** Assume $\sum_{i=1}^{n} \phi_1(x_i)c_i = \sum_{i=1}^{n} \phi_1(x_i)d_i$ for $n \in \mathbb{N}$, $c_i, d_i \in \mathcal{A}$. Since $\{\phi_1(x) \mid x \in \mathcal{X}\}$ is $\mathcal{A}$-linearly independent, we have $c_i = d_i$ for $i = 1, \ldots, n$. Thus, $P_f \sum_{i=1}^{n} \phi_1(x_i)c_i = \sum_{i=1}^{n} \phi_1(f(x_i))c_i = \sum_{i=1}^{n} \phi_1(f(x_i))d_i = P_f \sum_{i=1}^{n} \phi_1(x_i)d_i.$ $\square$

**Lemma 2.8**  *Let $k_1 = \tilde{k}a$, i.e., $k$ is separable, for an invertible operator $a$ and a complex-valued kernel $\tilde{k}$. Assume $\{\tilde{\phi}(x) \mid x \in \mathcal{X}\}$ is linearly independent (e.g. $\tilde{k}$ is Gaussian or Laplacian), where $\tilde{\phi}$ is the feature map associated with $\tilde{k}$. Then, $\{\phi_1(x) \mid x \in \mathcal{X}\}$ is $\mathcal{A}$-linearly independent.*

**Proof** If $\sum_{i=1}^{n} \phi_1(x_i)c_i = 0$, we have

$$0 = \left\langle \sum_{i=1}^{n} \phi_1(x_i)c_i, \sum_{i=1}^{n} \phi_1(x_i)c_i \right\rangle_{\mathcal{M}_k} = c^*Gc = (G^{1/2}c)^*(G^{1/2}c),$$

where $G$ is the $\mathcal{A}^{n \times n}$-valued Gram matrix whose $(i,j)$-entry is $k(x_i, x_j) \in \mathcal{A}$ and $c = (c_1, \ldots, c_n)$. Thus, we obtain $G^{1/2}c = 0$. Let $\tilde{G}$ be the standard Gram matrix whose $(i,j)$-entry is $\tilde{k}(x_i, x_j)$. Since $G = \tilde{G} \otimes a$ and $\tilde{G}$ is invertible, the inverse of $G$ is $\tilde{G}^{-1} \otimes a^{-1}$. Thus, $G^{1/2}$ is invertible and we have $c = 0$. $\square$

**Lemma 4.2**  *Let $\mathcal{F}$ be a function class of $\mathbb{R}^{d \times d}$-valued functions on $\mathcal{A}_0$ bounded by $C$ (i.e., $\|f(x)\|_{\mathcal{A}} \leq C$ for any $x \in \mathcal{A}_0$). Let $\tilde{\mathcal{G}}(\mathcal{F}, p) = \{(x,y) \mapsto \|(f(x)-y)p\|^2 \mid f \in \mathcal{F}, \|y\|_{\mathcal{A}} \leq E\}$ and $M = 2(C+E)^2$. Let $p \in \mathbb{R}^d$ satisfy $\|p\| = 1$ and let $\delta \in (0,1)$. Then, for any $g \in \tilde{\mathcal{G}}(\mathcal{F}, p)$, with probability at least $1 - \delta$, we have*

$$\|\mathrm{E}[|g(x,y)|_{\mathcal{A}}^2]^{1/2}p\|^2 \leq \left\| \frac{1}{n} \sum_{i=1}^{n} |g(x_i, y_i)|_{\mathcal{A}}^2 \right\|_{\mathcal{A}} + 2\hat{R}_n(\mathbf{x}, \tilde{\mathcal{G}}(\mathcal{F}, p)) + 3M\sqrt{\frac{\log(2/\delta)}{2n}}.$$

**Proof** For $(x, y), (x', y') \in \mathcal{A} \times \mathbb{R}^{d \times d}$, we have

$$\|(f(x)-y)p\|^2 - \|(f(x')-y')p\|^2$$
$$= \|f(x)p\|^2 - 2\langle f(x)p, yp\rangle + \|yp\|^2 - \|f(x')p\|^2 + 2\langle f(x')p, y'p\rangle - \|y'p\|^2 \leq 2(C+E)^2.$$

In addition, we have

$$\|\mathrm{E}[|g(x,y)|_{\mathcal{A}}^2]^{1/2}p\|^2 - \left\| \left( \frac{1}{n} \sum_{i=1}^{n} |g(x_i, y_i)|_{\mathcal{A}}^2 \right)^{1/2} p \right\|^2 = \langle p, \mathrm{E}[|g(x,y)|_{\mathcal{A}}^2]p\rangle - \left\langle p, \frac{1}{n} \sum_{i=1}^{n} |g(x_i, y_i)|_{\mathcal{A}}^2 p \right\rangle$$

$$= \mathrm{E}[\langle p, |g(x,y)|_{\mathcal{A}}^2 p\rangle] - \frac{1}{n} \sum_{i=1}^{n} \langle p, |g(x_i, y_i)|_{\mathcal{A}}^2 p\rangle = \mathrm{E}[\|g(x,y)p\|^2] - \frac{1}{n} \sum_{i=1}^{n} \|g(x_i, y_i)p\|^2.$$

Since $(x, y) \mapsto \|g(x, y)p\|^2$ is a real-valued map, by Theorem 3.3 by Mohri et al. [11], we have

$$\|\mathrm{E}[|g(x,y)|_{\mathcal{A}}^2]^{1/2}p\|^2 - \left\| \left( \frac{1}{n} \sum_{i=1}^{n} |g(x_i, y_i)|_{\mathcal{A}}^2 \right)^{1/2} p \right\|^2 \leq 2\hat{R}_n(\mathbf{x}, \tilde{\mathcal{G}}(\mathcal{F}, p)) + 3M\sqrt{\frac{\log(2/\delta)}{2n}}.$$

The inequality $\|(\sum_{i=1}^{n} |g(x_i, y_i)|_{\mathcal{A}}^2)^{1/2}p\|^2 \leq \|\sum_{i=1}^{n} |g(x_i, y_i)|_{\mathcal{A}}^2\|_{\mathcal{A}}$ completes the proof. $\square$

**Lemma 4.3** *With the same notations in Lemma 4.2, let $K = 2\sqrt{2}(C + E)$. Then, we have $\hat{R}_n(\mathbf{x}, \tilde{\mathcal{G}}(\mathcal{F}, p)) \leq K\hat{R}_n(\mathbf{x}, \mathcal{F}p)$, where $\mathcal{F}p = \{x \mapsto f(x)p \mid f \in \mathcal{F}\}$.*

**Proof** Let $h_i(z) = \|z - y_i p\|^2$ for $z \in \{f(x)p \mid f \in \mathcal{F}\} \subseteq \mathbb{R}^d$ and $i = 1, \ldots, n$. The Lipschitz constant of $h_i$ is calculated as follows: We have

$$
\begin{aligned}
h_i(z) - h_i(z') &= \|z\|^2 - 2\langle z - z', y_i p \rangle - \|z'\|^2 \\
&\leq (\|z\| + \|z'\|)(\|z\| - \|z'\|) + 2\|z - z'\|\|y_i p\| \\
&\leq (\|z\| + \|z'\|)(\|z - z'\| + \|z'\| - \|z'\|) + 2\|z - z'\|\|y_i p\| \\
&\leq (2C + 2E)\|z - z'\|.
\end{aligned}
$$

Thus, we have $|h_i(z) - h_i(z')| \leq 2(C + E)\|z - z'\|$. By Corollary 4 by Maurer [40], the statement is proved. $\qquad\square$

**Lemma 4.4** *Let $p \in \mathbb{R}^d$ satisfy $\|p\| = 1$. For $\mathcal{F}_1$ defined in Section 3, we have*

$$
\hat{R}_n(\mathbf{x}, \mathcal{F}_1 p) \leq \frac{B_1}{n}\Big(\sum_{i=1}^n \mathrm{tr}(k(x_i, x_i))\Big)^{1/2}.
$$

**Proof** We have

$$
\begin{aligned}
\mathrm{E}\Big[\sup_{f \in \mathcal{F}_1} \frac{1}{n}\sum_{i=1}^n \langle \sigma_i, f(x_i)p \rangle\Big] &= \mathrm{E}\Big[\sup_{f \in \mathcal{F}_1} \frac{1}{n}\sum_{i=1}^n \big\langle (\sigma_i p^*)p, \langle \phi_1(x_i), f \rangle_{\mathcal{M}_1} p \big\rangle\Big] \\
&= \mathrm{E}\Big[\sup_{f \in \mathcal{F}_1} \frac{1}{n}\Big\langle p, \sum_{i=1}^n (p\sigma_i^*)\langle \phi_1(x_i), f \rangle_{\mathcal{M}_1} p \Big\rangle\Big] \\
&= \mathrm{E}\Big[\sup_{f \in \mathcal{F}_1} \frac{1}{n}\Big\langle p, \Big\langle \sum_{i=1}^n \phi_1(x_i)(\sigma_i p^*), f \Big\rangle_{\tilde{\mathcal{M}}_1} p \Big\rangle\Big] \\
&= \mathrm{E}\Big[\sup_{f \in \mathcal{F}_1} \frac{1}{n}\Big(\sum_{i=1}^n \phi_1(x_i)(\sigma_i p^*), f\Big)_{\tilde{\mathcal{M}}_1, p}\Big] \\
&\leq \mathrm{E}\Big[\sup_{f \in \mathcal{F}_1} \frac{1}{n}\Big\|\sum_{i=1}^n \phi_1(x_i)(\sigma_i p^*)\Big\|_{\tilde{\mathcal{M}}_1, p} \|f\|_{\tilde{\mathcal{M}}_1, p}\Big] \quad &(3) \\
&\leq \mathrm{E}\Big[\sup_{f \in \mathcal{F}_1} \frac{1}{n}\Big\langle p, \Big|\sum_{i=1}^n \phi_1(x_i)(\sigma_i p^*)\Big|_{\tilde{\mathcal{M}}_1}^2 p \Big\rangle^{1/2}\|f\|_{\tilde{\mathcal{M}}_1}\Big] \quad &(4) \\
&\leq \frac{B_1}{n}\mathrm{E}\Big[\Big(\sum_{i,j=1}^n p^* p \sigma_i^* k_1(x_i, x_j)\sigma_j p^* p\Big)^{1/2}\Big] \\
&\leq \frac{B_1}{n}\mathrm{E}\Big[\sum_{i,j=1}^n \sigma_i^* k_1(x_i, x_j)\sigma_j\Big]^{1/2} \quad &(5) \\
&= \frac{B_1}{n}\mathrm{E}\Big[\sum_{i=1}^n \sigma_i^* k_1(x_i, x_i)\sigma_i\Big]^{1/2} = \frac{B_1}{n}\Big(\sum_{i=1}^n \mathrm{tr}\, k_1(x_i, x_i)\Big)^{1/2},
\end{aligned}
$$

where for $p \in \mathbb{C}^d$ and $f, g \in \tilde{\mathcal{M}}_1$, $(f, g)_{\tilde{\mathcal{M}}_1, p}$ is the semi-inner product defined by $(f, g)_{\tilde{\mathcal{M}}_1, p} = \langle p, \langle f, g \rangle_{\tilde{\mathcal{M}}_1} p \rangle$ and $|f|_{\tilde{\mathcal{M}}_1}^2 = \langle f, f \rangle_{\tilde{\mathcal{M}}_1}$. In addition, the inequality (3) is by the Cauchy–Schwartz inequality and the inequality (5) is by the Jensen's inequality. Note that the Cauchy–Schwartz inequality is still valid for semi-inner products. The inequality (4) is derived by the inequality

$$
\|f_L\|_{\tilde{\mathcal{M}}_L, p}^2 = \langle p, \langle f_L, f_L \rangle_{\mathcal{M}_L} p \rangle \leq \|p\|^2 \|\langle f_L, f_L \rangle_{\mathcal{M}_L}\|_{\mathcal{A}} \leq \|f_L\|_{\tilde{\mathcal{M}}_L}^2.
$$

$\square$

**Theorem 4.1** *Assume there exists $D > 0$ such that $\|k_1(x,x)\|_{\mathcal{A}} \leq D$ for any $x \in \mathcal{A}_0$. Let $\tilde{K} = 4\sqrt{2}(\sqrt{D}B_1 + E)B_1$ and $\tilde{M} = 6(\sqrt{D}B_1 + E)^2$. Let $\delta \in (0,1)$. Then, for any $g \in \mathcal{G}(\mathcal{F}_1)$, where $\mathcal{F}_1$ is defined in Section 3, with probability at least $1 - \delta$, we have*

$$\|\mathrm{E}[|g(x,y)|^2_{\mathcal{A}}]\|_{\mathcal{A}} \leq \left\| \frac{1}{n}\sum_{i=1}^n |g(x_i,y_i)|^2_{\mathcal{A}} \right\|_{\mathcal{A}} + \frac{\tilde{K}}{n}\left( \sum_{i=1}^n \mathrm{tr}\, k_1(x_i,x_i) \right)^{1/2} + \tilde{M}\sqrt{\frac{\log(2/\delta)}{2n}}.$$

**Proof** For $f \in \mathcal{M}_1$ and $x \in \mathcal{A}_0$, we have

$$\|f(x)\|_{\mathcal{A}} = \| \langle \phi_1(x), f \rangle_{\mathcal{M}_1} \|_{\mathcal{A}} \leq \|k_1(x,x)\|^{1/2}_{\mathcal{A}}\|f\|_{\mathcal{M}_1} \leq \sqrt{D}B_1.$$

Thus, we set $C$ as $\sqrt{D}B_1$ and apply Lemmas 4.2, 4.3, and 4.4. Then, for $p \in \mathbb{R}^d$ satisfy $\|p\| = 1$, with probability at least $1 - \delta$, we have

$$\|\mathrm{E}[|g(x,y)|^2_{\mathcal{A}}]^{1/2}p\|^2 \leq \left\| \frac{1}{n}\sum_{i=1}^n |g(x_i,y_i)|^2_{\mathcal{A}} \right\|_{\mathcal{A}} + \frac{\tilde{K}}{n}\left( \sum_{i=1}^n \mathrm{tr}\, k_1(x_i,x_i) \right)^{1/2} + \tilde{M}\sqrt{\frac{\log(2/\delta)}{2n}}.$$

Therefore, we obtain

$$\|\mathrm{E}[|g(x,y)|^2_{\mathcal{A}}]\|_{\mathcal{A}} = \|\mathrm{E}[|g(x,y)|^2_{\mathcal{A}}]^{1/2}\|^2_{\mathcal{A}}$$
$$\leq \left\| \frac{1}{n}\sum_{i=1}^n |g(x_i,y_i)|^2_{\mathcal{A}} \right\|_{\mathcal{A}} + \frac{\tilde{K}}{n}\left( \sum_{i=1}^n \mathrm{tr}\, k_1(x_i,x_i) \right)^{1/2} + \tilde{M}\sqrt{\frac{\log(2/\delta)}{2n}},$$

which completes the proof. $\square$

**Proposition 4.6** *We have*

$$\hat{R}_n(\mathbf{x}, \mathcal{F}_L^{\mathrm{deep}}p) \leq \frac{1}{n}\sup_{(f_j \in \mathcal{F}_j)_j} \|P_{f_{L-1}} \cdots P_{f_1}|_{\tilde{\mathcal{V}}(\mathbf{x})}\|_{\mathrm{op}} \|f_L\|_{\mathcal{M}_L} \left( \sum_{i=1}^n \mathrm{tr}(k_1(x_i,x_i)) \right)^{1/2}.$$

*Here, $\tilde{\mathcal{V}}(\mathbf{x})$ is the submodule of $\tilde{\mathcal{M}}_1$ generated by $\phi_1(x_1), \dots \phi_1(x_n)$.*

The following lemma by Lance [33, Proposition 1.2] is used in proving Proposition 4.6. Here, for $a, b \in \mathcal{A}$, $a \leq b$ means $b - a$ is Hermitian positive definite.

**Lemma A.1** *Let $\mathcal{M}$ and $\mathcal{N}$ be Hilbert $\mathcal{A}$-modules and let $A$ be an $\mathcal{A}$-linear operator from $\mathcal{M}$ to $\mathcal{N}$. Then, we have $|Aw|^2_{\mathcal{N}} \leq \|A\|^2_{\mathrm{op}}|w|^2_{\mathcal{M}}$.*

**Proof**

$$\mathrm{E}\left[ \sup_{f \in \mathcal{F}_L^{\mathrm{deep}}} \frac{1}{n}\sum_{i=1}^n \langle \sigma_i, f(x_i)p \rangle \right] = \mathrm{E}\left[ \sup_{(f_j \in \mathcal{F}_j)_j} \frac{1}{n}\sum_{i=1}^n \langle (\sigma_i p^*)p, f_L(f_{L-1}(\cdots f_1(x_i)\cdots))p \rangle \right]$$
$$= \mathrm{E}\left[ \sup_{(f_j \in \mathcal{F}_j)_j} \frac{1}{n}\sum_{i=1}^n \langle (\sigma_i p^*)p, \langle \phi_L(f_{L-1}(\cdots f_1(x_i)\cdots)), f_L \rangle_{\mathcal{M}_L} p \rangle \right]$$
$$= \mathrm{E}\left[ \sup_{(f_j \in \mathcal{F}_j)_j} \frac{1}{n}\sum_{i=1}^n \langle p, (p\sigma_i^*) \langle \phi_L(f_{L-1}(\cdots f_1(x_i)\cdots)), f_L \rangle_{\mathcal{M}_L} p \rangle \right]$$
$$= \mathrm{E}\left[ \sup_{(f_j \in \mathcal{F}_j)_j} \frac{1}{n} \left\langle p, \left\langle \sum_{i=1}^n \phi_L(f_{L-1}(\cdots f_1(x_i)\cdots))(\sigma_i p^*), f_L \right\rangle_{\tilde{\mathcal{M}}_L} p \right\rangle \right]$$

$$= \mathrm{E}\left[\sup_{(f_j\in\mathcal{F}_j)_j} \frac{1}{n}\left(\sum_{i=1}^n \phi_L(f_{L-1}(\cdots f_1(x_i)\cdots))(\sigma_i p^*), f_L\right)_{\tilde{\mathcal{M}}_{L,p}}\right]$$

$$\leq \mathrm{E}\left[\sup_{(f_j\in\mathcal{F}_j)_j} \frac{1}{n}\left\|\sum_{i=1}^n \phi_L(f_{L-1}(\cdots f_1(x_i)\cdots))(\sigma_i p^*)\right\|_{\tilde{\mathcal{M}}_{L,p}} \|f_L\|_{\tilde{\mathcal{M}}_{L,p}}\right] \qquad (6)$$

$$\leq \mathrm{E}\left[\sup_{(f_j\in\mathcal{F}_j)_j} \frac{1}{n}\left\|P_{f_{L-1}}\cdots P_{f_1}\sum_{i=1}^n \phi_1(x_i)(\sigma_i p^*)\right\|_{\tilde{\mathcal{M}}_{L,p}} \|f_L\|_{\tilde{\mathcal{M}}_L}\right]$$

$$= \mathrm{E}\left[\sup_{(f_j\in\mathcal{F}_j)_j} \frac{1}{n}\left\langle p, \left|P_{f_{L-1}}\cdots P_{f_1}\sum_{i=1}^n \phi_1(x_i)(\sigma_i p^*)\right|^2_{\tilde{\mathcal{M}}_L} p\right\rangle^{1/2} \|f_L\|_{\tilde{\mathcal{M}}_L}\right]$$

$$\leq \mathrm{E}\left[\sup_{(f_j\in\mathcal{F}_j)_j} \frac{1}{n}\left\langle p, \|P_{f_{L-1}}\cdots P_{f_1}|_{\tilde{\mathcal{V}}(\mathbf{x})}\|^2_{\mathrm{op}}\left|\sum_{i=1}^n \phi_1(x_i)(\sigma_i p^*)\right|^2_{\tilde{\mathcal{M}}_L} p\right\rangle^{1/2} \|f_L\|_{\tilde{\mathcal{M}}_L}\right] \quad (7)$$

$$\leq \frac{1}{n}\sup_{(f_j\in\mathcal{F}_j)_j} \|P_{f_{L-1}}\cdots P_{f_1}|_{\tilde{\mathcal{V}}(\mathbf{x})}\|_{\mathrm{op}} \|f_L\|_{\mathcal{M}_L}\,\mathrm{E}\left[\left\langle p, \left|\sum_{i=1}^n \phi_1(x_i)(\sigma_i p^*)\right|^2_{\tilde{\mathcal{M}}_L} p\right\rangle^{1/2}\right]$$

$$\leq \frac{1}{n}\sup_{(f_j\in\mathcal{F}_j)_j} \|P_{f_{L-1}}\cdots P_{f_1}|_{\tilde{\mathcal{V}}(\mathbf{x})}\|_{\mathrm{op}} \|f_L\|_{\mathcal{M}_L}\,\mathrm{E}\left[\left(\sum_{i,j=1}^n p^* p\sigma_i^* k_1(x_i,x_j)\sigma_j p^* p\right)^{1/2}\right]$$

$$\leq \frac{1}{n}\sup_{(f_j\in\mathcal{F}_j)_j} \|P_{f_1}\cdots P_{f_{L-1}}|_{\tilde{\mathcal{V}}(\mathbf{x})}\|_{\mathrm{op}} \|f_L\|_{\mathcal{M}_L}\left(\sum_{i=1}^n \mathrm{tr}(k_1(x_i,x_i))\right)^{1/2},$$

where the inequality (6) is by the Cauchy–Schwartz inequality and the inequality (7) is by Lemma A.1.
$\square$

**Proposition 5.1** *Let* $h : \mathcal{A}^n \times \mathcal{A}^n \to \mathbb{R}_+$ *be an error function, let* $g_1$ *be an* $\mathbb{R}_+$*-valued function on the space of bounded linear operators on* $\tilde{\mathcal{M}}_1$*, and let* $g_2 : \mathbb{R}_+ \to \mathbb{R}_+$ *satisfy* $g_2(a) \leq g_2(b)$ *for* $a \leq b$*. Assume the following minimization problem has a solution:*

$$\min_{(f_j\in\mathcal{M}_j)_j} h(f_L\circ\cdots\circ f_1(x_1),\ldots,f_L\circ\cdots\circ f_1(x_n)) + g_1(P_{f_{L-1}}\cdots P_{f_L}|_{\tilde{\mathcal{V}}(\mathbf{x})}) + g_2(\|f_L\|_{\mathcal{M}_L}).$$

*Then, there exists a solution admitting a representation of the form* $f_j = \sum_{i=1}^n \phi_j(x_i^{j-1})c_{i,j}$ *for some* $c_{1,j},\ldots,c_{n,j} \in \mathcal{A}$ *and for* $j = 1,\ldots,L$*. Here,* $x_i^j = f_j\circ\cdots\circ f_1(x_i)$ *for* $j = 1,\ldots,L$ *and* $x_i^0 = x_i$*.*

**Proof** Let $\mathcal{V}_j(\mathbf{x})$ be the submodule of $\mathcal{M}_j$ generated by $\phi_j(x_1^{j-1}),\ldots,\phi_j(x_n^{j-1})$. Let $f_j = f_j^{\|} + f_j^{\perp}$, where $f_j^{\|} \in \mathcal{V}_j(\mathbf{x})$ and $f_j^{\perp} \in \mathcal{V}_j(\mathbf{x})^{\perp}$. Then, for $j = 1,\ldots,L-1$, we have

$$f_j(x_i^{j-1}) = \langle\phi_j(x_i^{j-1}), f_j\rangle_{\mathcal{M}_j} = \langle\phi_j(x_i^{j-1}), f_j^{\|}\rangle_{\mathcal{M}_j} = f_j^{\|}(x_i^{j-1}). \qquad (8)$$

In addition, we have

$$P_{f_{L-1}}\cdots P_{f_1}|_{\tilde{\mathcal{V}}(\mathbf{x})} = \prod_{j=1}^{L-1} P_{f_j}|_{\tilde{\mathcal{V}}_j(\mathbf{x})},$$

where $\tilde{\mathcal{V}}_j(\mathbf{x})$ is the submodule of $\tilde{\mathcal{M}}_j$ generated by $\phi_j(x_1^{j-1}),\ldots\phi_j(x_n^{j-1})$. In the same manner as Eq. (8), we obtain

$$P_{f_j}\sum_{i=1}^n \phi_j(x_i^{j-1})c_{i,j} = \sum_{i=1}^n \phi_{j+1}(f_j(x_i^{j-1}))c_{i,j} = \sum_{i=1}^n \phi_{j+1}\big(f_j^{\|}(x_i^{j-1})\big)c_{i,j}$$

$$= \sum_{i=1}^n P_{f_j^{\|}}\phi_j(x_i^{j-1})c_{i,j}.$$

Thus, we have $P_{f_j}|_{\tilde{\mathcal{V}}_j(\mathbf{x})} = P_{f_j^{\parallel}}|_{\tilde{\mathcal{V}}_j(\mathbf{x})}$. Furthermore, we have

$$\|f_L\|_{\mathcal{M}_L} = \|\langle f_L^{\parallel} + f_L^{\perp}, f_L^{\parallel} + f_L^{\perp}\rangle_{\mathcal{M}_L}\|_{\mathcal{A}} = \|\langle f_L^{\parallel}, f_L^{\parallel}\rangle_{\mathcal{M}_L} + \langle f_L^{\perp}, f_L^{\perp}\rangle_{\mathcal{M}_L}\|_{\mathcal{A}} \geq \|\langle f_L^{\parallel}, f_L^{\parallel}\rangle_{\mathcal{M}_L}\|_{\mathcal{A}},$$

where the last inequality is derived from the fact that $\langle f_L^{\parallel}, f_L^{\parallel}\rangle_{\mathcal{M}_L} + \langle f_L^{\perp}, f_L^{\perp}\rangle_{\mathcal{M}_L} - \langle f_L^{\perp}, f_L^{\perp}\rangle_{\mathcal{M}_L}$ is positive and Theorem 2.2.5 (3) by Murphy [34]. As a result, the statement is proved. $\square$

**Proposition 5.3** *For $j = 1, L$, let $[\phi_j(x_1^{j-1}), \ldots, \phi_j(x_n^{j-1})]R_j = Q_j$ be the QR decomposition of $[\phi_j(x_1^{j-1}), \ldots, \phi_j(x_n^{j-1})]$. Then, we have $\|P_{f_{L-1}} \cdots P_{f_1}|_{\tilde{\mathcal{V}}(\mathbf{x})}\|_{\mathrm{op}} = \|R_L^* G_L R_1\|_{\mathrm{op}}$.*

**Proof** The result is derived by the identities

$$\begin{aligned}
\|P_{f_{L-1}} \cdots P_{f_1}|_{\tilde{\mathcal{V}}(\mathbf{x})}\|_{\mathrm{op}} &= \|Q_L^* P_{f_{L-1}} \cdots P_{f_1} Q_1\|_{\mathrm{op}} = \|Q_L^* P_{f_{L-1}} \cdots P_{f_1} [\phi_1(x_1), \ldots, \phi_1(x_n)]R_1\|_{\mathrm{op}} \\
&= \|R_L^* [\phi_L(x_1^{L-1}), \ldots, \phi_L(x_n^{L-1})]^* [\phi_L(x_1^{L-1}), \ldots, \phi_L(x_n^{L-1})]R_1\|_{\mathrm{op}} \\
&= \|R_L^* G_L R_1\|_{\mathrm{op}}.
\end{aligned}$$

$\square$

**Proposition 5.4** *Assume $G_L$ is invertible. Then, we have*
$$\|P_{f_{L-1}} \cdots P_{f_1}|_{\tilde{\mathcal{V}}(\mathbf{x})}\|_{\mathrm{op}} \leq \|G_L^{-1}\|_{\mathrm{op}}^{1/2} \|G_L\|_{\mathrm{op}} \|G_1^{-1}\|_{\mathrm{op}}^{1/2}.$$

**Proof** Since $G_L$ is invertible, by Proposition 5.3, $\|R_L^* G_L R_1\|_{\mathrm{op}}$ is bounded as

$$\|P_{f_{L-1}} \cdots P_{f_1}|_{\tilde{\mathcal{V}}(\mathbf{x})}\|_{\mathrm{op}} = \|R_L^* G_L R_1\|_{\mathrm{op}} \leq \|R_L\|_{\mathrm{op}} \|G_L\|_{\mathrm{op}} \|R_1\|_{\mathrm{op}} = \|G_L^{-1}\|_{\mathrm{op}}^{1/2} \|G_L\|_{\mathrm{op}} \|G_1^{-1}\|_{\mathrm{op}}^{1/2},$$

where the last equality is derived by the identity $\|R_j\|_{\mathrm{op}}^2 = \|R_j R_j^*\|_{\mathrm{op}} = \|G_j^{-1}\|_{\mathrm{op}}$. $\square$

# B  Connection with Generalization Bound with Koopman Operators

Hashimoto et al. [18] derived a generalization bound for the classical neural network composed of linear transformations and activation functions using Koopman operators. The Koopman operator is the adjoint of the Perron–Frobenius operator. In their analysis, they set the final transformation $f_L$ in an RKHS and observed that if the transformation of each layer is injective, the noise is separated through the linear transformations and cut off by $f_L$. Since the final transformation $f_L$ in our case is in an RKHM, the same interpretation about noise is valid for deep RKHM, too. Indeed, if $f_L \circ \cdots \circ f_1$ is not injective, then $G_L$ is not invertible, which results in the right-hand side of the inequality (2) going to infinity. The bound derived by Hashimoto et al. also goes to infinity if the network is not injective.

# C  Experimental Details and Additional Results

We provide details for the experiments in Section 7. All the experiments are executed with Python 3.9 and TensorFlow 2.6 on Intel(R) Core(TM) i9 CPU and NVIDIA Quadro RTX 5000 GPU with CUDA 11.7.

## C.1  Comparison with vvRKHS

We set $k_j(x, y) = \tilde{k}(x, y)I$ for $j = 1, \ldots, 3$ with $\tilde{k}(x, y) = \mathrm{e}^{-c \sum_{i,j=1}^d |x_{i,j} - y_{i,j}|}$ and $c = 0.001$ for positive definite kernels. For the optimizer, we used SGD. The learning rate is set as $1e^{-4}$ both for the deep RKHM and deep vvRKHS. The initial value of $c_{i,j}$ is set as $a_{i,j} + \epsilon_{i,j}$, where $a_{i,j} \in \mathcal{A}_j$ is the block matrix all of whose elements are 0.1 and $\epsilon_{i,j}$ is randomly drawn from $\mathcal{N}(0, 0.05)$. The result in Figure 2 is obtained by 5 independent runs.

## C.2 Observation about benign overfitting

We set the same positive definite kernel as Subsection C.1 for $j = 1, 2$. For the optimizer, we used SGD. The learning rate is set as $3 \times 1e^{-4}$. The initial value of $c_{i,j}$ is the same as Subsection C.1. The result in Figure 2 is obtained by 3 independent runs.

## C.3 Comparison to CNN

We set $k_j(x, y) = \tilde{k}(xa_j, ya_j)xy^*$ for $j = 1, 2$, where $a_1$ and $a_2$ are block matrices whose block sizes are 2 and 4, for positive definite kernels. All the nonzero elements of $a_j$ are set as 1. We set $a_1$ and $a_2$ to induce interactions in the block elements (see Remark 5.2). For the optimizer, we used Adam with learning rate $1e^{-3}$ for both the deep RKHM and the CNN. The initial value of $c_{i,j}$ is set as $\epsilon_{i,j}$, where $\epsilon_{i,j}$ is randomly drawn from $\mathcal{N}(0, 0.1)$.

We combined the deep RKHM and CNN with 2 dense layers. Their activation functions are sigmoid and softmax, respectively. For the CNN layers, we also used the sigmoid activation functions. The loss function is set as the categorical cross-entropy for the CNN. The result in Figure 2 is obtained by 5 independent runs.

### C.3.1 Memory consumption and computational cost

**Memory consumption**  We used a CNN with $(7 \times 7)$ and $(4 \times 4)$-filters. On the other hand, for the deep RKHM, we learned the coefficients $c_{i,j}$ in Proposition 5.1 for $j = 1, 2$ and $i = 1, \ldots, n$. That is, we learned the following coefficients:

- $n(= 20)$ block diagonal matrices each of whom has four $(7 \times 7)$-blocks (for the first layer)
- $n$ block diagonal matrices each of whom has seven $(4 \times 4)$-blocks (for the second layer)

Thus, the size of the parameters we have to learn for the deep RKHM is larger than the CNN. Since the memory consumption depends on the size of learning parameters, the memory consumption is larger for the deep RKHM than for the CNN.

**Computational cost**  In each iteration, we compute $f(x_i)$ for $i = 1, \ldots, n$ and the derivative of $f$ with respect to the learning parameters. Here, $f = f_1 \circ f_2$ is the network. For the deep RKHM, we compute the product of a Gram matrix (composed of $n \times n$ block diagonal matrices) and a vector (composed of $n$ block diagonal matrices) for computing $f_j(x_i)$ for all $i = 1, \ldots, n$. Thus, the computational cost for computing $f(x_i)$ for all $i = 1, \ldots, n$ is $O(n^2 d(m_1 + m_2))$, where $m_1 = 7$ and $m_2 = 4$ are the sizes of the block diagonal matrices. For the CNN, the computational cost for computing $f(x_i)$ for all $i = 1, \ldots, n$ is $O(nd^2(l_1 + l_2))$, where $l_1 = 7 \times 7$ and $l_2 = 4 \times 4$ are the number of elements in the filters. Since we set $n = 20$ and $d = 28$, the computational cost of the deep RKHM for computing $f(x_i)$ for $i = 1, \ldots, n$ is smaller than that of the CNN. However, since the size of the learning parameters of the deep RKHM is large, the computational cost for computing the derivative of $f(x_i)$ is larger than that of the CNN.

**Additional results**  To compare the deep RKHM to a CNN with the same size of learning parameters (the same memory consumption), we conducted the same experiment as the last experiment in the main text, excepting for the structure of the CNN. We constructed a CNN with $(28 \times 7 \cdot 20)$ and $(28 \times 4 \cdot 20)$-filters (The size of learning parameters is the same as the deep RKHM) and replaced the CNN used in the experiment in the main text. Figure 3 shows the result. The result is similar to that in Figure 2 (c), and the deep RKHM also outperforms the CNN with the same learning parameter size. Since the size of the learning parameter is the same in this case, the computational cost for one iteration for learning the deep RKHM is the same or smaller than that for learning the CNN.

### C.3.2 Additional results for benign ovefitting

We also observed benign overfitting for the deep RKHM. We compared the train and test losses for the deep RKHM with $\lambda_1 = 1$ to those with $\lambda_1 = 0$. The results are illustrated in Figure 4. If we set $\lambda_1 = 0$ in Eq. (1), i.e., do not minimize the term in Eq. (2), then whereas the training loss becomes small as the learning process proceeds, the test loss becomes large after sufficient iterations. On the other hand, if we set $\lambda_1 = 1$, i.e., minimize the term in Eq. (2), then the training loss becomes

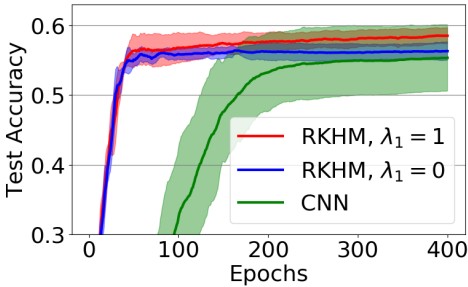

Figure 3: Comparison of deep RKHM to a CNN with the same size of learning parameter.

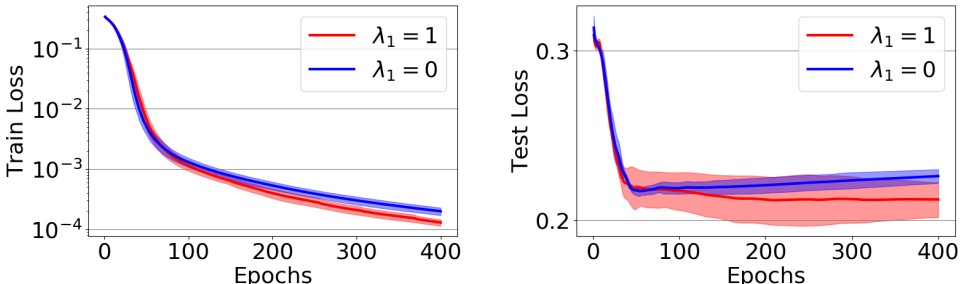

Figure 4: Train loss and test loss for the MNIST classification task with ($\lambda = 1$) and without ($\lambda = 0$) the minimization of the second term in Eq. (1) regarding the Perron–Frobenius operators.

smaller than the case of $\lambda_1 = 0$, and the test loss does not become large even the learning process proceeds. The result implies that the term regarding the Perron–Frobenius operators in Eq. (1) causes overfitting, but it is benign overfitting.

