# OpenReview forum: "Deep learning with kernels through RKHM and the Perron-Frobenius operator"
_NeurIPS.cc/2023/Conference — NeurIPS 2023 poster_

### Official Review · Reviewer_gE5P · 2023-07-06

**Soundness:** 4 excellent
**Presentation:** 4 excellent
**Contribution:** 4 excellent
**Rating:** 7
**Confidence:** 3

**Summary:**

The authors combines RKHM and Perron-Frobenious Operator to deep RKHM, a deep learning framework for kernel methods. By virtue of $C^*$-algebra, they manage to get a better bound on Rademacher generalization error and provide a clear connection with CNNs. Their theoretical analysis provides a new lens for deep kernel theory.

**Strengths:**

1. Clear writing
2. Very solid results
3. Novel tools
4. Their work can be inspiring.
5. Experiments support their theories

**Weaknesses:**

As the authors say, more efficient methods specific to deep RKHM remains to be investigated in future work. It remains a problem to scale up to at least ImageNet to be useful.

**Questions:**

Any theoretical understanding of why deep RKHMs might be better than nondeep ones?

**Limitations:**

The authors very adequately addressed the limitations. Very impressive.

---

> ### Author Rebuttal · Authors · 2023-08-07
>
> ### More efficient methods specific to deep RKHM
> As we stated in Rem. 5.6, we can apply random Fourier features to reduce the computational cost, but as you point out, methods specific to deep RKHM should be investigated as future work.
> We will study more about this topic and conduct some experiments with ImageNet to check the computational efficiency.
>
> ### Advantage of deep RKHM over shallow RKHM
> The motivation for studying deep kernel methods is that we try to combine the flexibility of deep neural networks with the representation power and solid theoretical understanding of kernel methods.
> From the theoretical point of view, the arguments in Subsection 6.2 about benign overfitting are valid only for deep RKHMs (i.e., $L\ge 2$).
> If $L=1$ (shallow RKHM), then the Gram matrix $G_L=[k(x_i,x_j)]$ is fixed and determined only by the training data and the kernel.
> On the other hand, if $L\ge 2$ (deep RKHM), then $G_L=[k_L(f_{L-1} \circ \cdots \circ f_1(x_i),f_{L-1} \circ \cdots \circ f_1(x_j))]$ depends also on $f_{1},\ldots,f_{L-1}$.
> As a result, by adopting the regularization using $G_L$, we can learn proper $f_1,\ldots,f_{L-1}$ so that they make the regularization term small and the whole network overfits benignly.
> As $L$ becomes large, the function $f_{L-1}\circ\cdots\circ f_1$ changes more flexibly to attain a smaller value of the regularization term.

---

> > ### Comment · Reviewer_gE5P · 2023-08-10
> >
> > makes sense.

---

### Official Review · Reviewer_HsSA · 2023-07-07

**Soundness:** 3 good
**Presentation:** 2 fair
**Contribution:** 2 fair
**Rating:** 6
**Confidence:** 3

**Summary:**

The authors introduce a generalization of RKHS for $C^*$ algebra valued kernels, called RKHM; they build networks by composing sequentially elements taken from a collection of RKHMs, one RKHM per layer. They prove generalization bounds for those networks. These networks output matrices at each layer.

**Strengths:**

I believe that a strength of the paper is that the generalization bound obtained in this paper for RKHM is better than the known ones for vvRKHS. It is unclear to me if RKHMs generalize vvRKHSs: maybe considering RKHM in the commutative $C^*$ of diagonal matrices is a way to represent a vvRKHS as an RKHM. If it is the case then the result of this paper is a better generalization bound for vvRKHS.

**Weaknesses:**

I feel that the paper is sometimes difficult to read. For example the name '$\mathcal{A}$-valued positive definite kernel' does not refer to the space $\mathcal{X}$ that characterizes the domain of the kernel; when defining deep RKHM knowing this information could help make explicit what the domain of $k_j$ as being $A_{j-1}\times A_{j-1}$.
Two remarks in this direction are on some notations:
- line 83:  shouldn't the content of the brace in the definition of $M_{k,0}$ be $\sum_{i=1}^{n} \phi(x_i) c_i \vert n\in \mathbb{N}, (c_i\in A, i\leq n), (x_i\in \mathcal{X}, i\leq n) $

- equation line 195, maybe the notation: $(f_j\in \mathcal{M}_j)_j$, could recall that the optimization is over all the RKHMs that define the networks.

**Questions:**

In 'Connection with neural tangent kernel', is the aim of this paragraph to define a neural tangent kernel for deep RKHM?

In 'Comparison to CNN', how long does the training take and what is the memory consumption of RKHM and CNN?

**Limitations:**

In Conclusion and Limitations, I feel that the statement 'connections with existing studies such as CNNs and neural tangent kernel' is a bit of a strong statement as the authors explain in section 6.1 that CNN and RKHM do not really relate to one another and it is unclear to me how deep the connection to neural tangent kernel is.

---

> ### Author Rebuttal · Authors · 2023-08-07
>
> ### Relation between RKHM and vvRKHS
> RKHM is a generalization of vvRKHS in the sense that we can reconstruct vvRKHS using RKHM (Please refer Thm. 4.13 of [2]).
> Since the output space of the functions in a vvRKHS is not necessarily a $C^*$-algebra, the connection between RKHM and vvRKHS is a little complicated.
> An important advantage of RKHM over vvRKHS is that we can incorporate the flexibility of $C^*$-algebra into kernel methods.
> Indeed, the idea of using the operator norm to improve the bound proposed in this paper comes from the perspective of RKHM, and cannot be obtained from the perspective of vvRKHS.
>
> As for the generalization bound, if the kernel has the form $k=\tilde{k}I$ for a complex-valued kernel $\tilde{k}$ and the $D\times D$ identity matrix $I$, then our result reduces the dependency on the output dimension of the vvRKHS.
> This is achieved by putting $D$ elements in a $D$-dimensional vector $v$ in a $\lceil \sqrt{D}\rceil\times \lceil \sqrt{D}\rceil$ matrix.
> Then, instead of considering $D\times D$ matrix-valued kernel for the vvRKHS, we only need $\lceil \sqrt{D}\rceil\times \lceil \sqrt{D}\rceil$ matrix-valued kernel for the RKHM.
> As a result, the dependency reduces from $D$ to $\lceil \sqrt{D}\rceil$.
> Since the action of the above kernel to a vector is described by the elementwise multiplication of the complex-valued kernel $\tilde{k}$ to the vector, the construction of the matrix-valued kernel is trivial.
> We can generalize the above argument to the case of $k=\tilde{k}a$ for general $a\in\mathbb{C}^{D\times D}$ by considering $av$ instead of $v$ to construct the $\lceil \sqrt{D}\rceil\times \lceil \sqrt{D}\rceil$ matrix.
> Since separable kernels are widely used, our result is valid in many cases.
> In more general cases, the relationship between RKHMs and vvRKHSs is more complicated, and investigating how we can apply this type of argument to vvRKHS with more general kernels is future work.
>
> ### Readability
> We will improve readability based on your comments.
>
> ### Connection with neural tangent kernel
> The aim of this paragraph is to define a neural tangent kernel for $C^*$-algebra network (a generalization of neural networks by means of $C^*$-algebra, please see [3]) and developing a theory for combining neural networks and deep RKHMs.
> Existing studies investigate the connection between neural networks and kernel methods through neural tangent kernels.
> We show the analogy of the existing connection for $C^*$-algebra networks and RKHMs.
> As we stated in Rem. 6.2, by virtue of the arguments in this paragraph, we can regard deep neural networks as one-layer (shallow) RKHM, which enables us to analyze the combination of neural networks and deep RKHMs.
>
> ### Comparison to CNN
> **Memory consumption**
> In the experiment of the comparison to CNNs in the paper, we used a CNN with $(7\times 7)$ and $(4\times 4)$-filters.
> On the other hand, for the deep RKHM we used in this experiment, we learned the coefficients $c_{i,j}$ in Prop. 5.1 for $j=1,2$ and $i=1,\ldots,n$.
> That is, we learned the following coefficient:
>
> - $n(=20)$ block diagonal matrices each of whom has four $(7\times 7)$-blocks (for the 1st layer)
>
> - $n$ block diagonal matrices each of whom has seven $(4\times 4)$-blocks (for the 2nd layer)
>
> Thus, the size of the parameters we have to learn for the deep RKHM is larger than the CNN.
> Since the memory consumption depends on the size of learning parameters, the memory consumption is larger for the deep RKHM than for the CNN.
>
> **Computational cost**
> In each iteration, we compute $f(x_i)$ for $i=1,\ldots,n$ and the derivative of $f$ with respect to the learning parameters.
> Here, $f=f_1\circ f_2$ is the network.
> For the deep RKHM, we compute the product of a Gram matrix (composed of $n\times n$ block diagonal matrices) and a vector (composed of $n$ block diagonal matrices) for computing $f_j(x_i)$ for all $i=1,\ldots,n$.
> Thus, the computational cost for computing $f(x_i)$ for all $i=1,\ldots,n$ is $O(n^2d(m_1+m_2))$, where $m_1=7$ and $m_2=4$ are the sizes of the block diagonal matrices.
> For the CNN, the computational cost for computing $f(x_i)$ for all $i=1,\ldots,n$ is $O(nd^2(l_1+l_2))$, where $l_1=7\times 7$ and $l_2=4\times 4$ are the number of elements in the filters.
> Since we set $n=20$ and $d=28$, the computational cost of the deep RKHM for computing $f(x_i)$ for $i=1,\ldots,n$ is smaller than that of the CNN.
> However, since the size of the learning parameters of the deep RKHM is large, the computational cost for computing the derivative of $f(x_i)$ is larger than that of the CNN.
>
> **Additional experiment**
> To compare the deep RKHM to a CNN with the same size of learning parameters (the same memory consumption), we did an additional experiment.
> We constructed a CNN with $(28\times 7\cdot 20)$ and $(28\times 4\cdot 20)$-filters (The size of learning parameters is the same as the deep RKHM) and replaced the CNN used in the paper with this CNN.
> Please see Fig. 2 in the PDF file attached at the top of the review part.
> The result is similar to that in the paper, and the deep RKHM also outperforms the CNN with the same learning parameter size.
> Since the size of the learning parameter is the same in this case, the computational cost for one iteration for learning the deep RKHM is the same or smaller than that for learning the CNN.
>
> ### Connection with existing studies
> In our paper, the term "connection" does not mean that the two objects are the same.
> In Sec. 6.1, we showed that whereas CNNs learn filters, deep RKHMs learn activation functions.
> As for the neural tangent kernel, as we also stated in the response of the first question, the arguments in Sec. 6.3 enable us to analyze the combination of neural and kernel networks.
> This makes our analysis more flexible.
> This paper is the first paper on deep kernel methods with RKHM, and we provide new features for deep kernel methods.
> We will investigate connections with existing methods deeper in future work.

---

### Official Review · Reviewer_Hurc · 2023-07-09

**Soundness:** 3 good
**Presentation:** 3 good
**Contribution:** 3 good
**Rating:** 7
**Confidence:** 3

**Summary:**

This paper proposes deep Reproducing kernel Hilbert $\mathcal{C}^*$-module (RKHM), a deep learning framework for kernel methods， which generalizes RKHS by means of $\mathcal{C}^*$-algebra. In this setting, a map as the composition of functions in RKHMs is constructed. Theoretically, the authors develop a new Rademacher generalization bound of deep RKHM using Perron-Frobenious norm. The dependency of the bound on the output dimension is milder than existing bounds. Moreover, they show a representer theorem to guarantee that the solution of a given practical minimization problem is represented only with given data. In addition, they show connections of deep RKHM with existing studies such as CNNs, benign overfitting and neural tangent kernel. Furthermore, this paper presents a series of numerical experiments to support their theory and show the practical performance of deep RKHM.


**Strengths:**

This paper provides a new approach to analyze and understand deep kernel methods. In particular, the authors derives a generalization bound for deep RKHMs, while existing work focuses on shallow RKHMs. This bound also relaxes the dependence on the output dimension by using Perron-Frobenious norm. It is also interesting how this bound relates to benign overfitting.

Moreover, this paper provides some experiments to support their theoretical findings.

The paper is technically sound and the contents are very relevant to the community of NeurIPS.

**Weaknesses:**

The paper is well-organized and well-written.

**Questions:**

Can you explain how the generalization bounds in Theorem 4.1 and Theorem 4.5 depend on the input dimension?

**Limitations:**

The authors have clearly addressed the limitations of this work.

---

> ### Author Rebuttal · Authors · 2023-08-07
>
> ### Dependency on the input dimension
> The input dimension depends on the generalization bound through the $\mathcal{A}$-valued kernel $k$.
> In Thms. 4.1 and 4.5, the bound depends on $\mathrm{tr}(k(x_i,x_i))$.
> Therefore, the dependency of the input dimension on the bound is determined by the dependency of the input dimension on the $\mathcal{A}$-valued kernel $k$.

---

### Official Review · Reviewer_eQG4 · 2023-07-10

**Soundness:** 3 good
**Presentation:** 3 good
**Contribution:** 2 fair
**Rating:** 5
**Confidence:** 3

**Summary:**

The paper establishes properties on the composition of functions belonging to a RKHM, a generaliztion of RKHSs. The authors compute the Rademacher complexity of this function class and establish a representer theorem.

**Strengths:**

- The objects studied in the paper are well introduced. The writing is generally clear.
- The paper adds another way formalizing the composition of linear models.

**Weaknesses:**

- The derivations of both the results of the paper are standard : the computation of Rademacher complexity using Cauchy-Schwartz + Jensen is essentially the same for all linear models. It is reproduced here with more terminology and higher dimensions but the steps are the same. The representer theorem which relies simply on the fact that the orthogonal component doesn't affect the objective is the same standard proof for kernels.
- It is unclear what is gained from this additional abstraction. The paragraph on the connection to CNNs is unconvincing: CNNs learn the filters, whereas (composition) of linear models have fixed filters but learn the coefficients of the terms in the sum. It is difficult to argue that this much abstraction is necessary to derive this observation.

    *The additional abstraction leads to doubts over existence of the objects*:  the existence of a "well defined" Perron-Frobenius operator is only established for a very specific case in Lemma 2.8, which appears to be artificial as it is a regular complex valued kernel multiplied by a matrix to create a multi-output kernel.

- The term benign overfitting does not seem appropriate as it is discussed in section 6.2 of the paper :  The term benign overfitting refers to the phenomenon observed that *complex function classes* can fit training data noise without loosing performance on the test set. In section 6.2 the discussion is on the *complexity of the function class*. The authors say that *regularization*  in eq(2) with the operator norm of the Perron-Frobenius operator composition decreases the function class complexity and therefore leads to better generalization, which is normal and expected. This goes completely against the unusual empirical observations that led to the study of benign overfitting - which is that *complex function classes generalize without regularization*. How can benign overfitting be discussed by analyzing a **uniform** generalization bound ?

- The general motivation of this work can be further developed (i.e slightly longer first paragraph) : Why should we formalize compositions of linear models ? The representer theorem is good to have but we have no convexity, so why use a composition of linear models instead of simply learning the feature embeddings as well if convexity is already lost ? Why are "deep" kernels worthy of study ?

**Questions:**

- What exactly is being done when writing "Thus" in the proof of Lemma 2.7 ? What does well defined mean? Are you showing existence of such a map ? It is unclear.
- How is Corollary 4.7 derived ? Can you give more details. In particular, how do you control the operator norm of the Perron-Frobenius operator without assuming that the intermediate $\phi_i$ are bounded ?

**Limitations:**

Limitations are discussed.

---

> ### Author Rebuttal · Authors · 2023-08-07
>
> ### Derivations of the results
> Our Rademacher complexity analysis and representer theorem for RKHM use Cauchy-Schwarz, Jensen inequalities, and orthogonality arguments, as most of the results in the RKHS case. However, it is important to be precise that there are specificities and technical difficulties peculiar to RKHM that we adequately addressed. This is why we improved the bound.
> Our main idea is to use the operator norm to reduce the dependency of the generalization bound on the output dimension.
> In the existing study of RKHM, the trace of operators is considered instead of the operator norm.
> Availability of the operator norm is by virtue of the application of $C^*$-algebra, and whether we can combine the standard approaches, such as Cauchy-Schwartz and Jensen, with the operator norm or not is nontrivial.
> For example, in the proofs of Lem. 4.4 and Prop. 4.6, we used a semi-inner product.
> Also, in the proof of Prop. 4.6, we need another lemma (Lem. A.1) to split the operator norm of the Perron-Frobenius (P-F) operator $P_{f_{L-1}}\cdots P_{f_1}$ and the absolute value of $\sum_{i=1}^n\phi_1(x_i)(\sigma_ip^*)$.
> Applying P-F operators is also our main idea and not a standard approach.
>
> ### Connection with CNN and well-definedness of the Perron-Frobenius (P-F) operators
> Our idea to connect kernel methods and CNN is based on the concepts of $C^*$-algebra and deep RKHM. Our connection relies on the observation that by considering the $C^*$-algebra $\mathcal{A}$ of circulant matrices, we can represent the convolution as the product of elements in the $C^*$-algebra.
> Indeed, $\mathcal{A}$-valued positive definite kernel makes the output of each layer become a circulant matrix, which enables us to apply the convolution as the product of the output and a parameter.
> We mentioned in the paper that for deep RKHM, we learn the coefficients $c_i$, while for CNN we learn the filters $a_j$ (line 244-245). It seems reasonable to interpret this difference as a consequence of solving the problem in the primal or in the dual. In the primal, the number of the learning parameters depends on the dimension of the data (or the filter for convolution), while in the dual, it depends on the size of the data.
>
> Moreover, separable kernels ($\mathcal{A}$-valued kernel defined by a complex-valued kernel and an operator) are widely used in existing literature of vvRKHS; please see e.g. [1].
> We might well say that it is the most used class of operator-valued kernels in the ML literature. It includes all scalar-valued kernels as a special case.
> Lem. 2.8 guarantees the validity of the analysis with Perron-Frobenius operators, at least in the separable case.
> This paper is the first paper that combines deep kernel methods with $C^*$-algebra.
> Our study provides a condition for "good kernels" by means of the well-definedness of the P-F operator.
> As we already mentioned in Conclusion of our paper, more detailed analysis for other kernels is future work.
>
> ### Benign overfitting
> We are sorry the term "regularization" may be misleading since the operator norm of the P-F composition depends on the training data.
> The 2nd term in Eq. (1) and the right hand side of Eq. (2) are expected to cause overfitting since these terms depend on the training data. (The Gram matrix $G_L$ depends on the data.)
> Standard existing regularizations, such as the one for kernel ridge regression (the 3rd term in Eq. (1)) and $l_p$ regularization of weight matrices for neural networks, do not depend on the data.
> Please note the 3rd term in Eq. (1) does not depend on the training data before applying the representer theorem.
> The term in Eq. (2) is different from these standard regularizations.
>
> If we try to minimize a value depending on the training data, the model seems to be more specific for the training data, and it may cause overfitting.
> Thus, the connection between the minimization of the 2nd term in Eq. (1) and generalization cannot be explained by the classical argument about generalization and regularization, and we need the argument of benign overfitting.
> In the 3rd experiment in Sec. 7, we can also observe benign overfitting.
> Please see Fig. 1 in the PDF file attached at the top of the review part.
> If $\lambda_1=0$ in Eq. (1) (do not minimize the term in Eq. (2)), then whereas the training loss becomes small as the learning process proceeds, the test loss becomes large after sufficient iterations.
> On the other hand, if $\lambda_1=1$ (minimize the term in Eq. (2)), then the training loss becomes smaller than the case of $\lambda_1=0$, and the test loss does not become large even the learning process proceeds.
>
> ### Motivation
> Let us first note the models we are considering are nonlinear.
> All of the functions $f_1,\ldots,f_L$ introduced in Sec. 3 are nonlinear functions.
> By virtue of the composition, the model $f=f_L\circ \cdots\circ f_1$ is also nonlinear with respect to the coefficients $c_{i,j}$ introduced in the representer thm (Prop. 5.1).
>
> The motivation for studying deep kernel methods is that we try to combine the flexibility of deep neural networks with the representation power and solid theoretical understanding of kernel methods.
> We cannot guarantee the convexity of the loss functions, but the situation is the same as the case of neural networks.
> We don't have the convexity of the loss functions for neural networks.
> Our motivation for studying deep RKHM is that we apply the novel perspective of RKHM to deep kernel methods in order to make the deep kernel methods more powerful.
>
> ### Answer of the Questions
> - Well-definedness of $P_f$ means "if $u=v$ for $u,v\in M_{k,0}$, then $P_fu=P_fv$". We are sorry the last formula in the proof of Lem. 2.7 should be $P_f\sum_{i=1}^n\phi_1(x_i)d_i$ (it is a typo).
> - Cor. 4.7 is derived by applying the inequalities $\Vert P_{f_j}\Vert\le B_j$ ($j=1,\ldots,L-1$) and $\Vert f_L\Vert\le B_L$ to the result in Prop. 4.6. These inequalities for intermediate layers are by the definition of $\mathcal{F}_j$.

---

### Author Rebuttal · Authors · 2023-08-07

## To all the reviewers
Thank you very much for your constructive comments. We address your questions and concerns below. We will revise our paper based on your comments and the response below for the camera-ready version.

We attached a PDF file to support some of our responses here. Also, we provide references for our responses here.

[1] Mauricio A.  Alvarez, Lorenzo Rosasco, Neil D. Lawrence, et al. Kernels for vector-valued functions: A review. Foundations and Trends in Machine Learning, 4(3):195–266, 2012.

[2] Yuka Hashimoto, Isao Ishikawa, Masahiro Ikeda, Fuyuta Komura, Takeshi Katsura, and Yoshinobu Kawahara. Reproducing kernel Hilbert $C^*$-module and kernel mean embeddings. JMLR, 22(267):1–56, 2021.

[3] Ryuichiro Hataya and Yuka Hashimoto. Noncommutative $C^*$-algebra net: Learning neural networks with powerful product structure in $C^*$-algebra. arXiv: 2302.01191.

---

### Decision · Program_Chairs · 2023-09-21

**Decision:**

Accept (poster)

**Comment:**

This paper proposes deep Reproducing kernel Hilbert-module (RKHM), a deep learning framework for kernel methods, which generalizes RKHSs. The authors compute the Rademacher complexity of this function class and establish a representer theorem.

All reviewers liked the paper and therefore my recommendation is to accept it.